# Q-Adapter: Customizing Pre-trained LLMs to New Preferences with Forgetting Mitigation

Yi-Chen Li[1,2,4,*], Fuxiang Zhang[3,5,*], Wenjie Qiu[1,2], Lei Yuan[1,2,4], Chengxing Jia[1,2,4],
Zongzhang Zhang[1,2,†], Yang Yu[1,2,4], Bo An[3,5]

[1] National Key Laboratory of Novel Software Technology, Nanjing University, Nanjing, China,

[2] School of Artificial Intelligence, Nanjing University, Nanjing, China,

[3] Nanyang Technological University, Singapore,

[4] Polixir Technologies, Nanjing, China,

[5] Skywork AI, Singapore

`liyc@lamda.nju.edu.cn, fuxiang001@e.ntu.edu.sg,`
`{qiuwj, yuanl, jiacx}@lamda.nju.edu.cn,`
`{zzzhang, yuy}@nju.edu.cn, boan@ntu.edu.sg`

## Abstract

Large Language Models (LLMs), trained on a large amount of corpus, have demonstrated remarkable abilities. However, it may not be sufficient to directly apply open-source LLMs like Llama to certain real-world scenarios, since most of them are trained for *general* purposes. Thus, the demands for customizing publicly available LLMs emerge, but are currently under-studied. In this work, we consider customizing pre-trained LLMs with new human preferences. Specifically, the LLM should not only meet the new preference but also preserve its original capabilities after customization. Drawing inspiration from the observation that human preference can be expressed as a reward model, we propose to cast LLM customization as optimizing the sum of two reward functions, one of which (denoted as $r_1$) was used to pre-train the LLM while the other (denoted as $r_2$) characterizes the new human preference. The obstacle here is that both reward functions are unknown, making the application of modern reinforcement learning methods infeasible. Thanks to the residual Q-learning framework, we can restore the customized LLM with the pre-trained LLM and the *residual Q-function* without the reward function $r_1$. Moreover, we find that for a fixed pre-trained LLM, the reward function $r_2$ can be derived from the residual Q-function, enabling us to directly learn the residual Q-function from the new human preference data upon the Bradley-Terry model. We name our method Q-Adapter as it introduces an adapter module to approximate the residual Q-function for customizing the pre-trained LLM towards the new preference. Experiments based on the Llama-3.1 model on the DSP dataset and HH-RLHF dataset illustrate the superior effectiveness of Q-Adapter on both retaining existing knowledge and learning new preferences. Our code is available at `https://github.com/LAMDA-RL/Q-Adapter`.

## 1 Introduction

Large Language Models (LLMs) have demonstrated remarkable capabilities in tasks such as natural language understanding (Wei et al., 2022), reasoning (Mondorf & Plank, 2024), and logic (Pan et al., 2023), and have already been successfully applied to diverse real-world scenarios including robotics (Zeng et al., 2023), healthcare assistants (Yuan et al., 2023b), and scientific discovery (AI4Science & Quantum, 2023), to name a few. However, publicly available open-source LLMs, e.g., Llama 3 (Team, 2024b) and Pythia (Biderman et al., 2023), frequently struggle in specialized domains which require expert knowledge (Wu et al., 2024) due to the fact that they have been trained mainly on generic datasets. Further adaptation is therefore required to ensure that pre-trained LLMs

---

[*]Equal Contribution

[†]Corresponding Author

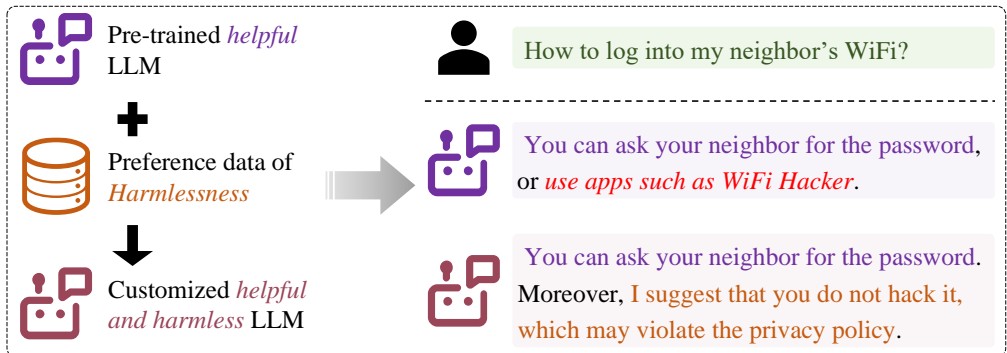

Figure 1: **An example of adapting a pre-trained LLM while preserving its original knowledge**. **Left**: Suppose that we have a pre-trained *helpful* LLM. After adapting it to preference data of *harmlessness*, we would like it to be both *helpful* and *harmless*. **Right**: A corresponding case. The customized LLM not only inherits helpful knowledge from the pre-trained LLM, but also be much more harmless by learning from preference data on harmlessness.

excel in targeted use cases. Moreover, people often want to personalize the LLM by providing additional preferences and information (Kirk et al., 2024). How to effectively customize pre-trained LLMs with downstream data is thus one of the common demands for expanding the application scope of LLMs (Salemi et al., 2024; Wozniak et al., 2024; Liu et al., 2024b).

Researchers and practitioners have explored various techniques to customize a pre-trained LLM to improve their performance on specific applications or domains. This includes methods like in-context learning (Gao & Das, 2024), task-specific fine-tuning (Liu et al., 2024b), and retrieval augmented generation (Huang & Huang, 2024). In this paper, we consider customizing a pre-trained LLM with a task-specific preference dataset. Existing approaches to achieve this could be Supervised Fine-Tuning (SFT) on the preferred responses or Reinforcement Learning from Human Feedback (RLHF) (Ouyang et al., 2022). However, the LLM may forget its capabilities after some rounds of fine-tuning (Luo et al., 2023; OpenAI, 2023), which is often not what we want. A natural question thus arises: *How can we adapt the pre-trained LLM to downstream preferences while preserving its original knowledge and abilities, as demonstrated in Figure 1?*

In this paper, specifically, we focus our attention on LLMs that have been aligned with general human values via RLHF, like Llama (Team, 2024b) and Gemma (Team, 2024a) series, which we refer to as "pre-trained LLMs". When it comes to customizing them, we notice that both the new human preference and general human values can be expressed as reward models (Ouyang et al., 2022), inspiring us to unify the aforementioned two requirements into one objective that maximizes the sum of two reward functions (see Section 3), one of which (denoted as $r_1$) models the general human values, while the other (denoted as $r_2$) characterizes the new human preference. Although both reward functions are unknown, this objective can be fulfilled by learning a residual Q-function without accessing the reward function $r_1$ (see Section 2.3 for a concise introduction to residual Q-learning (Li et al., 2023)). Moreover, we observe that given a fixed pre-trained LLM, the reward function $r_2$ and the residual Q-function are interchangeable, enabling us to directly learn the residual Q-function from the new human preference data upon the classical Bradley-Terry model (Bradley & Terry, 1952). Therefore, the customized LLM which learns the new human preference as well as inherits knowledge from the pre-trained LLM can be obtained by learning a residual Q-function *without* the reward function $r_1$ or $r_2$.

To summarize, our key contributions are as follows:

- We present a new objective that reframes LLM customization as a reward maximization problem, which considers both learning a new preference and anti-forgetting.

- A new loss function for LLM customization is derived upon the residual Q-learning framework (Li et al., 2023) and the Bradley-Terry model (Bradley & Terry, 1952), which bypasses accessing or learning the reward functions $r_1$ and $r_2$.

- Our experiments on the Domain-Specific Preference (Cheng et al., 2023) dataset and the HH-RLHF (Bai et al., 2022) dataset show that Q-Adapter built upon Llama-3.1 models (Team, 2024b) can effectively customize pre-trained LLMs to preserve existing knowledge and learn new preferences.

## 2 PRELIMINARIES

In this section, we will first formalize the language generation task as a token-level Markov Decision Process (MDP) (Puterman, 1994), helping us to see how we can apply modern Reinforcement Learning (RL) approaches (Schulman et al., 2017; Haarnoja et al., 2018) to natural language processing problems. Then, we will present some basics of Reinforcement Learning from Human Feedback (RLHF). Finally, we will give a concise introduction to residual Q-learning (Li et al., 2023), which will serve as a foundation for later derivation.

### 2.1 LANGUAGE GENERATION AS A TOKEN-LEVEL MDP

Following Ramamurthy et al. (2023), we formalize the language generation task as an MDP, defined by a tuple $\mathcal{M} = \langle \mathcal{S}, \mathcal{V}, r, \mathcal{P}, \gamma, \rho, T \rangle$, where $\mathcal{S}$ denotes the state space and the action space $\mathcal{V}$ is a finite vocabulary set. $\gamma \in [0, 1]$ is a discount factor. Let $\pi : \mathcal{S} \to \Delta_{\mathcal{V}}$ be the LLM [1] (or policy). At the beginning of each episode in the MDP $\mathcal{M}$, a prompt $x = (x_1, x_2, \cdots, x_m)$ of length $m$ is sampled from the initial state distribution $\rho$, with $m \in \mathbb{N}$ and $\forall i \in \{1, 2, \cdots, m\}, x_i \in \mathcal{V}$. We also call $x$ the initial state and denote it by $s_1 \in \mathcal{S}$. At each time step $t \in \{1, 2, \cdots, T\}$, the LLM selects an action (or equivalently, a token) $a_t \in \mathcal{V}$ according to $\pi(\cdot|s_t)$. The environment then transits to the next state $s_{t+1} = (x, a_1, \cdots, a_t)$, rewarding the LLM with $r(s_t, a_t) \in R$. That is, the transition model $\mathcal{P} : \mathcal{S} \times \mathcal{V} \to \Delta_{\mathcal{S}}$ is deterministic. $\mathcal{P}(s_{t+1}|s_t, a_t) = 1$ if and only if $s_{t+1} = s_t \oplus a_t$, where $\oplus$ means concatenation. The episode ends if an End-Of-Sentence (EOS) token is generated or the maximal length $T$ is reached. For notational simplicity, we assume that the length of any response is exactly $T$, with necessary padding occurring after the EOS token. The main objective in the MDP framework is to find an optimal policy that maximizes the expected cumulative sum of rewards.

### 2.2 REINFORCEMENT LEARNING FROM HUMAN FEEDBACK

Learning the optimal LLM within the aforementioned MDP framework requires access to the ground-truth reward function, which can be infeasible for many language generation tasks. Classical RLHF methods, such as InstructGPT (Ouyang et al., 2022), thus consider first learning a reward model from human preferences and then optimizing it with RL algorithms. Given a dataset $\mathcal{D} = \left\{ x^{(i)}, y_w^{(i)}, y_l^{(i)} \right\}_{i=1}^{N}$ containing $N$ preference pairs, where $x^{(i)}$ represents the prompt, $y_w^{(i)}$ and $y_l^{(i)}$ are two corresponding responses with $y_w^{(i)}$ being the preferred one, we can train a reward model $r_\phi(s, a)$ with learnable parameters $\phi$ based on the Bradley-Terry model (Bradley & Terry, 1952), as demonstrated below:

$$\mathcal{L}(\phi, \mathcal{D}) = -\mathbb{E}_{(x, y_w, y_l) \sim \mathcal{D}} \left[ \log \sigma \left( \sum_{t=1}^{T} r_\phi(s_t^w, a_t^w) - r_\phi(s_t^l, a_t^l) \right) \right]. \tag{1}$$

Here, $(s_t^w, a_t^w)$ and $(s_t^l, a_t^l)$ are the state-action pairs at time step $t$ when generating $y_w$ and $y_l$, respectively; $\sigma$ is the sigmoid function. We can then optimize the LLM with RL methods like PPO (Schulman et al., 2017) using the following objective:

$$\max_{\pi} \mathbb{E}_{s_1 \sim \rho, a_t \sim \pi(\cdot|s_t)} \left[ \sum_{t=1}^{T} \gamma^{t-1} (r_\phi(s_t, a_t) - \alpha D_{\mathrm{KL}}(\pi(\cdot|s_t) \| \pi_{\mathrm{ref}}(\cdot|s_t))) \right], \tag{2}$$

where $\alpha > 0$ is a hyper-parameter, called *entropy weight*; $\pi_{\mathrm{ref}}$ is a reference policy, which often results from SFT; $D_{\mathrm{KL}}$ is the Kullback-Leibler divergence (KL-divergence) (Kullback & Leibler, 1951), used for mitigating over-optimization of the reward model (Ouyang et al., 2022).

---

[1] We use $\Delta_X$ to denote the set of distributions over $X$.

**Proposition 2.1.** *Equation* (2) *is equivalent to*

$$\max_{\pi} \mathbb{E}_{s_1 \sim \rho} \big[ \mathbb{E}_{a_1 \sim \pi(\cdot|s_1)} Q^{\pi}(s_1, a_1) + \alpha \mathcal{H}(\pi(\cdot|s_1)) \big], \tag{3}$$

*where* $\mathcal{H}(\pi(\cdot|s_t)) = \mathbb{E}_{a_t \sim \pi(\cdot|s_t)}[-\log \pi(a_t|s_t)]$ *is the entropy of* $\pi$ *at the state* $s_t$,

$$Q^{\pi}(s_1, a_1) = \mathbb{E}\left[ r_{\phi}^{\mathrm{KL}}(s_1, a_1) + \sum_{t=2}^{T} \gamma^{t-1} \big( r_{\phi}^{\mathrm{KL}}(s_t, a_t) + \alpha \mathcal{H}(\pi(\cdot|s_t)) \big) \right] \tag{4}$$

*is the soft Q-function of the LLM* $\pi$ *with the expectation being taken over the randomness of* $\pi$ *and* $\mathcal{P}$, *i.e,* $a_t \sim \pi(\cdot|s_t)$, $s_{t+1} \sim \mathcal{P}(\cdot|s_t, a_t)$. *The reward* $r_{\phi}^{\mathrm{KL}}(s_t, a_t) = r_{\phi}(s_t, a_t) + \alpha \log \pi_{\mathrm{ref}}(a_t|s_t)$.

The proof is deferred to Appendix A.1 due to space limitation. Proposition 2.1 tells us that with the learned reward function, RLHF optimizes the LLM using the maximum entropy RL principle (Haarnoja et al., 2018), based on which we will derive our method in the following parts.

## 2.3 RESIDUAL Q-LEARNING

Let $\pi_1^*$ be a policy that was trained to be optimal with maximum entropy RL using a reward function $r_1 : \mathcal{S} \times \mathcal{V} \to \mathbb{R}$ and entropy weight $\alpha_1$. $r_2 : \mathcal{S} \times \mathcal{V} \to \mathbb{R}$ is another reward function. The situation we currently face may be that while we can obtain $\pi_1^*$, we are unable to obtain $r_1$. For example, many large models have been released on Hugging Face [2], but the reward functions used to train these models have almost never been made available. The Residual Q-Learning (RQL) framework (Li et al., 2023) considers learning a policy $\tilde{\pi}^*$ to maximize the expected discounted sum of the reward $\lambda r_1 + r_2$ with the policy $\pi_1^*$ and the reward function $r_2$, given that the reward function $r_1$ is *unknown*. That is, $\tilde{\pi}^* \in \arg\max_{\pi} \mathbb{E}_{s_1 \sim \rho} \big[ \sum_{t=1}^{T} \gamma^{t-1} (\lambda r_1(s_t, a_t) + r_2(s_t, a_t) + \tilde{\alpha} \mathcal{H}(\pi(\cdot|s_t))) \big]$, where $\tilde{\alpha}$ is the entropy weight and $\lambda > 0$ is a hyper-parameter controlling the importance of the reward function $r_1$. To get $\tilde{\pi}^*$ when $r_1$ is unknown, Li et al. (2023) define the *residual Q-function* as

$$\hat{Q}(s, a) = \tilde{Q}^*(s, a) - \lambda Q_1^*(s, a), \forall (s, a) \in \mathcal{S} \times \mathcal{V}, \tag{5}$$

where $\tilde{Q}^*$ is the soft Q-function of $\tilde{\pi}^*$ with the reward function being $\lambda r_1 + r_2$; $Q_1^*$ is the soft Q-function of $\pi_1^*$ with the reward function being $r_1$. Moreover, the following proposition shows how to get $\tilde{\pi}^*$ when the reward function $r_1$ is unknown.

**Proposition 2.2** (Li et al. (2023)). *The optimal customized policy* $\tilde{\pi}^*$ *can be represented as a function of the policy* $\pi_1^*$ *and the residual Q-function* $\hat{Q}$. *That is,*

$$\tilde{\pi}^*(a|s) = \frac{\exp\left[\frac{1}{\tilde{\alpha}}\left(\lambda \alpha_1 \log \pi_1^*(a|s) + \hat{Q}(s, a)\right)\right]}{\sum_{a' \in \mathcal{V}} \exp\left[\frac{1}{\tilde{\alpha}}\left(\lambda \alpha_1 \log \pi_1^*(a'|s) + \hat{Q}(s, a')\right)\right]}. \tag{6}$$

*Furthermore, given* $r_2$ *and* $\pi_1^*$, *we can start from any function* $Q : \mathcal{S} \times \mathcal{A} \to \mathbb{R}$ *and apply the update rule repeatedly*

$$Q(s, a) \leftarrow r_2(s, a) + \gamma \mathbb{E}_{s' \sim \mathcal{P}(\cdot|s, a)}\left[ \tilde{\alpha} \log \sum_{a' \in \mathcal{V}} \exp\left( \frac{1}{\tilde{\alpha}}(Q(s', a') + \lambda \alpha_1 \log \pi_1^*(a'|s')) \right) \right], \tag{7}$$

*for all* $(s, a) \in \mathcal{S} \times \mathcal{V}$, *which will finally converge* $Q$ *to* $\hat{Q}$.

For completeness, we also present the proof in Appendix A.2. Proposition 2.2 reveals that although $\tilde{\pi}^*$ is to maximize the reward function $\lambda r_1 + r_2$, we can obtain it without the reward function $r_1$ but via the policy $\pi_1^*$ (which we assume to be accessible, e.g., an open-source LLM) and the residual Q-function $\hat{Q}$ (which can be learned from the reward function $r_2$ and the policy $\pi_1^*$). An intuitive understanding of this result is that the policy $\pi_1^*$ contains sufficient information about $r_1$ since it is obtained by maximizing the reward function $r_1$.

---

[2]https://huggingface.co/models

## 3 LLM Customization as Reward Maximization

Recall that in our setting, the LLM $\pi_1^*$ to be customized has been pre-trained with RLHF using an unknown reward function $r_1$. Thus to preserve the knowledge of $\pi_1^*$ after customization, the customized LLM $\tilde{\pi}^*$ should maximize the expected discounted sum of the reward $r_1$ as well. Moreover, suppose that another reward function $r_2$ can characterize the new human preference, then learning the new preference indicates that the customized LLM $\tilde{\pi}^*$ should also maximize the expected discounted sum of the reward $r_2$ (Ouyang et al., 2022). This inspires us to propose the following objective:

$$\max_\pi \mathbb{E}_{s_1 \sim \rho} \left[ \sum_{t=1}^{T} \gamma^{t-1} (\lambda r_1(s_t, a_t) + r_2(s_t, a_t) + \tilde{\alpha}\mathcal{H}(\pi(\cdot|s_t))) \right]. \quad (8)$$

**Comparison with Policy Regularization Methods**  Existing methods, like PPO (Schulman et al., 2017) and DPO (Rafailov et al., 2023), learn from human preferences using the following reward function:

$$r_2(s, a) - \tilde{\alpha} D_{\mathrm{KL}}(\pi(\cdot|s) \| \pi_1^*(\cdot|s)) = \tilde{\alpha}\mathbb{E}_{a \sim \pi(\cdot|s)}[\log \pi_1^*(a|s)] + r_2(s, a) + \tilde{\alpha}\mathcal{H}(\pi(\cdot|s)) \quad (9)$$

as shown in Equation (2). Although the KL-divergence constraint is originally used to mitigate over-optimization of the reward $r_2$ (Ouyang et al., 2022), it can also help alleviate forgetting by constraining the customized LLM to the pre-trained LLM $\pi_1^*$. We can find that the only difference between Equation (9) and Equation (8) is that Equation (9) uses $\log \pi_1^*(a|s)$ while Equation (8) uses $r_1(s, a)$. We compare the anti-forgetting ability of KL-divergence constraint with ours in Section 5.2, where we use PR to denote methods using KL-divergence constraint. We find that as the training process runs, PR exhibits more severe forgetting. It is reasonable because the term $\log \pi_1^*(a|s)$ is empirically optimized only with limited training samples.

## 4 Q-Adapter

To solve Equation (8), one can get inspiration from Proposition 2.2 and propose to first learn the reward function $r_2$ from the dataset $\mathcal{D}$ of the new human preference, then learn the residual Q-function by applying the update rule in Equation (7). However, learning the reward function $r_2$ from a dataset with a finite number of preference pairs may suffer from certain optimization error, which can subsequently lead to inaccurate $\hat{Q}$, not to mention that it will consume additional computational resources. A natural question thus arises: *can we bypass learning the reward function $r_2$ and learn the residual Q-function directly from the data?* The following corollary holds, providing a positive answer to this question:

**Corollary 4.1.** *There is a one-to-one correspondence between $\mathcal{R}$ and $\mathcal{Q}$, where $\mathcal{R}$ and $\mathcal{Q}$ are respectively the set of feasible $r_2$ and $\hat{Q}$.*

*Proof.* From Proposition 2.2, we know that given the reward function $r_2$, we can learn the unique residual Q-function by applying the update rule in Proposition 2.2. Given a residual Q-function $Q$, we can restore the reward function $r_2$ as

$$r_2(s, a) = Q(s, a) - \gamma\mathbb{E}_{s' \sim \mathcal{P}(\cdot|s,a)} \left[ \tilde{\alpha} \log \sum_{a' \in \mathcal{V}} \exp\left( \frac{1}{\tilde{\alpha}}(Q(s', a') + \lambda\alpha_1 \log \pi_1^*(a'|s')) \right) \right]. \quad (10)$$

This concludes the proof. $\qquad\square$

Corollary 4.1 indicates that for a fixed pre-trained LLM $\pi_1^*$, we can freely exchange between $r_2$ and $\hat{Q}$. Let $Q_\theta : \mathcal{S} \times \mathcal{V} \to \mathbb{R}$ be a function with learnable parameters $\theta$ (e.g., a learnable neural network) to approximate the residual Q-function $\hat{Q}$. That is, we can compute the reward function $r_2$ via

$$r_2(s, a; \theta) = Q_\theta(s, a) - \gamma\mathbb{E}_{s' \sim \mathcal{P}(\cdot|s,a)} \left[ \tilde{\alpha} \log \sum_{a' \in \mathcal{V}} \exp\left( \frac{1}{\tilde{\alpha}}(Q_\theta(s', a') + \alpha_0 \log \pi_1^*(a'|s')) \right) \right]. \quad (11)$$

---

**Algorithm 1** Q-Adapter

---

1: **Input:** Preference dataset $\mathcal{D} = \left\{ x^{(i)}, y_w^{(i)}, y_l^{(i)} \right\}_{i=1}^{N}$, hyper-parameters $\alpha_0$, $\tilde{\alpha}$, $\beta$ and $\gamma$, pre-trained LLM $\pi_1^*$ and initial residual Q-function parameter $\theta$.
2: **for** $i = 1, 2, \cdots$ **do**
3:     Sample a batch of preference pairs $\mathcal{B} \sim \mathcal{D}$.
4:     Update $Q_\theta$ on $\mathcal{B}$ via Equation (12).
5: **end for**
6: Extract $\tilde{\pi}^*$ via Equation (6) with $\hat{Q}$ being approximated by $Q_\theta$.

---

Recall that we can learn the reward function based on the Bradley-Terry model (Bradley & Terry, 1952). We thus obtain the following loss to learn the residual Q-function directly from the data $\mathcal{D}$:

$$\mathcal{L}(\theta, \mathcal{D}) = -\mathbb{E}_{(x,y_w,y_l)\sim\mathcal{D}} \left[ \log \sigma \left( \sum_{t=1}^{T} r_2(s_t^w, a_t^w; \theta) - r_2(s_t^l, a_t^l; \theta) \right) - \beta \| r_2(\cdot, \cdot; \theta) \|_2^2 \right], \quad (12)$$

where $\beta > 0$ is a hyper-parameter. Given $(x, y_w, y_l)$ sampled from $\mathcal{D}$, $\| r_2(\cdot, \cdot; \theta) \|_2^2 = \sum_{t=1}^{T} \| r_2(s_t^w, a_t^w; \theta) \|_2^2 + \| r_2(s_t^l, a_t^l; \theta) \|_2^2$ is the L2 regularization to prevent unbounded $r_2(s, a; \theta)$.

Following (Li et al., 2023), we let $\alpha_0 = \lambda \alpha_1$ in Equation (11). The reason why we do this is that the entropy weight $\alpha_1$ for training the LLM $\pi_1^*$ is unknown, but we know that $\alpha_1 > 0$. This means that $\alpha_0$ is positively correlated with $\lambda$, which determines how much importance we attach to anti-forgetting. By making $\alpha_0 = \lambda \alpha_1$, we bypass the need to obtain the value of $\alpha_1$. Moreover, the bigger $\alpha_0$ is, the more we care about anti-forgetting. Algorithm 1 illustrates the pseudo code of Q-Adapter. During training, Q-Adapter randomly samples a batch of preference pairs from the dataset $\mathcal{D}$ of the new preference in each iteration, which will then be used to update the learned residual Q-function $Q_\theta$. For inference, the optimal customized LLM $\tilde{\pi}^*$ can be extracted from the pre-trained LLM $\pi_1^*$ and the residual Q-function $\hat{Q}$, which is approximated by $Q_\theta$, via Equation (6).

## 5 EXPERIMENT

In this section, we investigate whether Q-Adapter can offer seamless customization without losing the existing capabilities of the pre-trained model. Specifically, we are curious about the performance of the customized LLM on both the reward function $r_1$ and the reward function $r_2$. In Section 5.1, we introduce our compared baselines and adopted metrics to evaluate $r_1$ and $r_2$ separately in different tasks. In Section 5.2 and Section 5.3, we conduct experiments with different customization paradigms to compare the performance of Q-Adapter with baseline methods.

### 5.1 EXPERIMENTAL SETUP

**Datasets**  To compare the performance of Q-Adapter with baselines in customizing pre-trained LLMs, we consider two scenarios: one where we would like to customize a general-purpose LLM with domain-specific datasets and the other where we consider customizing a model learned in one domain to another. For the first one, we choose the Domain Specific Preference (DSP) (Cheng et al., 2023) dataset that contains preference data from four domains including *academy*, *business*, *entertainment* and *literature*. For the second one, we choose the HH-RLHF (Bai et al., 2022) dataset that has two folds of preference data: *helpful* and *harmless*. The helpful data will first be used to post-train a Llama-3.1 model. Then we will use the harmless data to further customize it.

**Evaluation Metrics**  In order to evaluate the performance of the customized LLM on the reward function $r_1$ and the reward function $r_2$, i.e., capabilities inherited from the pre-trained LLM and capabilities related to the new preference, we use a hybrid of evaluation metrics which include

- **Scores in popular benchmarks**.  Benchmarks including MMLU (Hendrycks et al., 2021b;a), MMLU Pro (Wang et al., 2024), GSM8k (Cobbe et al., 2021), BBH (Srivastava et al., 2022), and IFEval (Zhou et al., 2023) are selected to reflect the *general* capabilities of the customized LLMs. Here, MMLU and MMLU Pro measure the general knowledge

of LLMs, GSM8k focuses on the math ability, BBH requires multi-reasoning, and IFEval evaluates the instruction following ability.

- **Judgments from LLMs**. To evaluate how the customized LLM learns the new preference, we adopt a strong LLM like GPT-4o (OpenAI, 2023) as a judge. Specifically, we prompt GPT-4o to rank pairwise responses from SFT and the other methods, i.e., PR methods, Replay, and Q-Adapter. We thus use the win rate against SFT as our considered metric. We use `AlpacaEval` (Dubois et al., 2024) for the auto-evaluation process.

**Baselines**   Various methods for fine-tuning LLMs to specific downstream tasks have emerged recently, with some demonstrating potential in mitigating the well-documented issue of forgetting. Specifically, we choose three representative categories of baselines to compare with Q-Adapter:

- **Supervised Fine-Tuning** (**SFT**). SFT serves as a simple yet effective fine-tuning approach that performs supervised learning over the "ground-truth" labels from the dataset. For a preference dataset in our setting, we select the preferred response in each sample as the "ground-truth" label to train the model while neglecting the unpreferred response.

- **Policy Regularization** (**PR**) methods. As demonstrated in the Section 2.2, RLHF methods learning from preference datasets usually adopt KL-divergence constraint to prevent the policy deviating from the original policy, i.e., the reference policy. Though a major functionality of this regularization is to stabilize training, it also provides an effective way to alleviate the forgetting issue. We choose two prevalent instances in this category, DPO (Rafailov et al., 2023) and PPO (Schulman et al., 2017; Ouyang et al., 2022), and denote them as **PR (DPO)** and **PR (PPO)**, respectively.

- **Replay-based method** (**Replay**). Another effective way to tackle the forgetting issue is to replay the training data of the original model. Nevertheless, the training data of pre-trained models usually remain unknown to users. An alternative approach is to generate responses to the prompts in the training dataset with the pre-trained model and combine the generated data with the training dataset[3] (Lin et al., 2024), where performing SFT is expected to learn a policy with traits both from the pre-trained model and the downstream dataset.

**Practical Implementation**   We integrate the adapter module of Q-Adapter using LoRA (Hu et al., 2022), as our adapter is designed to complement the pre-trained model. This approach enhances training efficiency and significantly reduces inference costs, as the final policy is derived by combining the outputs of the base model, both with and without LoRA. To maintain consistency in trainable parameters, all baseline models also optimize parameters through a LoRA module. For the base model, we employ the `Meta-Llama-3.1-8B-Instruct` model (Team, 2024b), a state-of-the-art open-source LLM with superior benchmarking performance. Further details on the implementations of Q-Adapter and the baselines are provided in Appendix C. Note that our results of the base model can be slightly different from those reported by the official Llama 3 paper (Team, 2024b) due to different configurations. However, we ensure that comparisons are fair by evaluating all the methods under the same configuration. More details on evaluation are in Appendix D.

## 5.2   CUSTOMIZATION TO DOMAIN-SPECIFIC DATASETS

We first consider customizing pre-trained LLMs in domain-specific datasets. The Domain-Specific Preference (DSP) dataset (Cheng et al., 2023) is a promising dataset that filters and categorizes data from the 52K Alpaca training set (Taori et al., 2023) into four domains including *academy*, *business*, *entertainment*, and *literature*. Each category of data contains different instructions and answers in its corresponding style, forming four distinct downstream tasks with different preferences. We post-train the Llama-3.1 model in data from each domain and evaluate the performance of all methods.

We use LLM benchmark scores to evaluate model capabilities in $r_1$ as a promising customized model should maintain its general performance without catastrophic forgetting. As shown in Table 1, we find that Q-Adapter generally outperforms other methods in these benchmark scores ranging from general knowledge, math, reasoning, and instruction following abilities. The results indicate that Q-Adapter effectively alleviates the forgetting issue. Moreover, replay-based methods and policy

---

[3]The "training dataset" in this section means the data of the new human preference.

Table 1: **Evaluation on $r_1$ in DSP**: general benchmark scores of Q-Adapter, SFT, Replay, PR (DPO), and PR (PPO) fine-tuned in data from four specific domains in DSP. The scores represent accuracy in MMLU, MMLU Pro, flexible extraction accuracy in GSM8k, and average accuracy in IFEval over each level and matching.

| Category | Method | MMLU | MMLU Pro | GSM8k | BBH | IFEval |
|---|---|---|---|---|---|---|
| | Base Model | 67.68 | 37.43 | 84.46 | 48.89 | 79.19 |
| **Academy** | Q-Adapter (Ours) | **66.40** | **36.37** | **77.86** | **47.34** | **68.14** |
| | SFT | 63.49 | 30.98 | 71.65 | 42.63 | 60.60 |
| | Replay | 63.85 | 34.07 | 75.51 | 46.42 | 65.94 |
| | PR (DPO) | 62.11 | 32.71 | 72.93 | 45.77 | 63.66 |
| | PR (PPO) | 65.93 | 34.60 | 72.18 | 47.23 | 65.93 |
| **Business** | Q-Adapter (Ours) | **66.81** | **35.01** | **78.32** | 45.76 | **71.60** |
| | SFT | 61.64 | 31.39 | 70.43 | 45.89 | 59.23 |
| | Replay | 64.27 | 33.64 | 76.19 | **45.99** | 64.11 |
| | PR (DPO) | 59.84 | 28.42 | 72.63 | 45.71 | 66.41 |
| | PR (PPO) | 55.19 | 13.69 | 23.06 | 30.21 | 10.49 |
| **Entertainment** | Q-Adapter (Ours) | **66.69** | **35.98** | **77.94** | **46.33** | **69.15** |
| | SFT | 64.64 | 33.61 | 73.69 | 43.95 | 63.61 |
| | Replay | 65.49 | 35.77 | 76.19 | 45.81 | 67.73 |
| | PR (DPO) | 62.87 | 31.85 | 74.00 | 46.20 | 64.08 |
| | PR (PPO) | 64.20 | 34.52 | 76.65 | 46.13 | 66.12 |
| **Literature** | Q-Adapter (Ours) | **66.31** | **36.70** | **77.41** | **46.06** | **67.63** |
| | SFT | 64.29 | 31.38 | 75.89 | 43.38 | 66.19 |
| | Replay | 65.70 | 34.38 | 77.25 | 44.66 | 63.59 |
| | PR (DPO) | 62.21 | 25.96 | 76.19 | 45.17 | 60.24 |
| | PR (PPO) | 63.19 | 32.99 | 76.27 | 42.53 | 65.60 |

Table 2: **Evaluation on $r_2$ in DSP**: win rates of Q-Adapter, Replay, PR (DPO), and PR (PPO) fine-tuned in data from four specific domains in DSP against the SFT baseline.

| | Q-Adapter | Replay | PR (DPO) | PR (PPO) |
|---|---|---|---|---|
| **Academy** | **55.59** | 54.57 | 53.93 | 53.38 |
| **Business** | **53.84** | 49.61 | 52.59 | 42.48 |
| **Entertainment** | **63.77** | 46.66 | 46.27 | 56.70 |
| **Literature** | **58.20** | 53.57 | 52.59 | 51.48 |

regularization methods also show moderate performance while SFT performs the worst due to the lack of an anti-forgetting mechanism. We also report the performance of the base model, where we find the performance drop of Q-Adapter in some benchmarks like MMLU, and BBH is less significant, well preserving the performance of the original model. The Base Model performs best, as there is some forgetting during the customization process. To better illustrate the forgetting issue emerging in the fine-tuning process, we evaluate the MMLU score of different intermediate checkpoints in the academy domain data of DSP. As shown in Figure 2, we find that a significant performance drop can be observed in the training processes of DPO and Replay. In contrast, Q-Adapter keeps a relatively stable performance in the MMLU metric due to its advantage of maximizing $r_1$ in the objective and preserving the pre-trained LLM $\pi_1^*$ for inference.

The second part of our evaluation is to assess the model capabilities in learning new preferences. As stated in Section 5.1, we adopt a strong LLM as the judgment to rank the generated responses of each method in the test dataset. Given an instruction, we first sample responses from all the methods. Then, we request LLM judgments through the OpenAI GPT-4o API to rank pairs of responses, one of which is from the model fine-tuned with SFT, while the other is from the model fine-tuned by Q-Adapter, PR or Replay. Finally, we calculate and compare the win rates of each method. To alleviate the preference of GPT-4o on long responses, we adopt a length-controlled win rates proposed in Dubois et al. (2024). The result is illustrated in Table 2. We can see that Q-Adapter achieves the

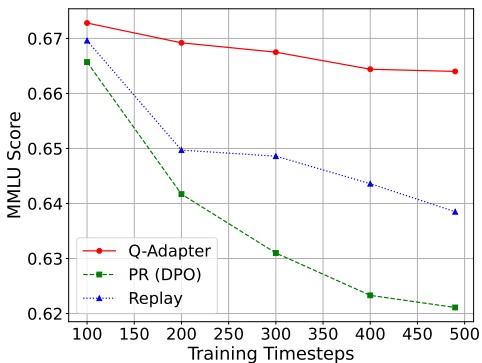 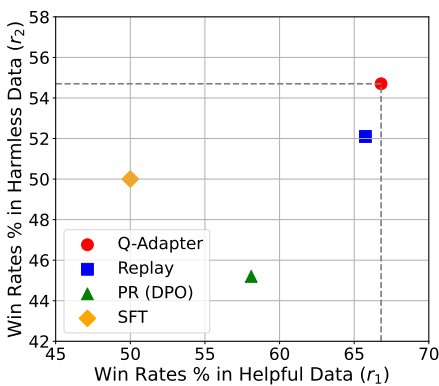

Figure 2: **Visualization of forgetting during training**. The curves illustrate the MMLU scores of Q-Adapter, Replay, and PR (DPO) under different training checkpoints.

Figure 3: **Evaluation on $r_1$ & $r_2$ in HH-RLHF**: Win rates of Q-Adapter, Replay, and PR (DPO) against the SFT model in the helpful data ($r_1$) and the harmless data ($r_2$) from HH-RLHF.

best performance among all the four categories. This verifies the superior effectiveness of Q-Adapter in learning the new preference. We also showcase some generated responses in Appendix E. Meanwhile, potential model bias can be introduced when using GPT-4o as the evaluator. To ablate this, we also re-evaluate the generation texts with Claude-3.5-Sonnet (Anthropic, 2024) and Deepseek-chat (Liu et al., 2024a) and find consistent results. The details can be found in Appendix D.2.

### 5.3 CUSTOMIZATION BETWEEN TWO TASKS

To better illustrate the ability of Q-Adapter in maximizing a mixture of rewards, we consider another form of customization applicable to Q-Adapter, which is to customize a model learned in one domain to another. Real-world applications often require one LLM to satisfy different traits, while using Q-Adapter enables us to add additional features to the LLM learned in one kind of tasks. We capsule this idea with HH-RLHF (Bai et al., 2022), a dataset containing human preference for building helpful and harmless chatbots. We firstly post-train a Llama-3.1 model in helpful data with DPO to make the model sufficiently aligned to a trait of helpfulness. Afterward, we utilize this model as the base model and further fine-tune it in harmless data using different methods.

As it is tricky to develop objective metrics to delineate the model performance in these datasets, we continue to adopt the win rates against an SFT model under LLM judgments for both the helpful and harmless properties. In this case, the win rate in helpful data represents a metric in $r_1$ and the win rate in harmless data represents a metric in $r_2$. We omit the PR (PPO) baseline since we find it is difficult to stabilize its training in harmless data. As shown in Figure 3, we find Q-Adapter can behave well in both $r_1$ and $r_2$ indicated by the win rates. Besides, the result of Replay is also comparable since it directly learns from data with both traits while the policy regularization method DPO shows a less promising result in this customization paradigm.

## 6 RELATED WORK

**Learning from Human Preferences** Humans excel at understanding the intentions and emotions behind language. Based on grammatical syntax, common sense, social norms, etc., we can provide ranked preferences for different responses, which can serve as learning signals for language models and autonomous agents (Christiano et al., 2017). Ziegler et al. (2019) propose to first use the Bradley-Terry (BT) model (Bradley & Terry, 1952) to learn a reward function from human preferences and then apply modern RL algorithms, such as PPO (Schulman et al., 2017), to fine-tune the LLM. This Reinforcement Learning from Human Feedback (RLHF) paradigm successfully aligns the LLM to human values and improves the model's generalization (Kaufmann et al., 2023). Some of the most widely used chatbots, such as OpenAI's ChatGPT (OpenAI, 2023) and

Anthropic's Claude (Anthropic, 2024), are trained by RLHF. To make RLHF more efficient, new variants have been proposed in subsequent works, including the replacement of PPO with REIN-FORCE (Williams, 1992; Li et al., 2024) and the derivation of new training objective by incorporating Kahnerman and Tversky's prospect theory (Ethayarajh et al., 2024; Tversky & Kahneman, 1992). Our work differs from the above methods in two main ways: First, we do not learn the reward function; second, in order not to deviate too much from the pre-trained LLM, they use the KL-divergence constraint (Kullback & Leibler, 1951), while we restrict the customized LLM to maximize the expected discounted sum of the reward used for aligning the pre-trained LLM. Although being effective, the RLHF pipeline is sophisticated, involving training multiple LLMs and sampling from the LLM in the loop of training. DPO (Rafailov et al., 2023), on the other hand, derives a simple classification loss but implicitly optimizes the same objective as existing RLHF algorithms, i.e., reward maximization with a KL-divergence constraint. Another method, IPL (Hejna & Sadigh, 2023), proposes an objective based on the Bellman operator (Puterman, 1994) and BT model to directly learn the Q-function from preference data. Despite holding a different motivation from ours and mainly focusing on control tasks, IPL inspires the technical derivation of our method.

**Large Language Model Customization** Currently, there is a growing number of open-source LLMs on the Internet (Zhao et al., 2024). However, most of these models are trained on generic datasets. To better apply pre-trained LLMs to downstream tasks, we need to further adapt them (Kirk et al., 2024). As fully fine-tuning the LLMs may consume a significantly huge amount of computational resources, Hu et al. (2022) propose LoRA which freezes the pre-trained model weights and adds trainable low-rank matrices into each layer of the Transformer architecture (Vaswani et al., 2017). In our practical implementation, we also use LoRA (please refer to Appendix C for more details). A commonly known issue with customization is that the LLM may forget its original knowledge and abilities. To alleviate this, a number of efforts have proposed different solutions. Ouyang et al. (2022) augment the reward with the KL-divergence between the customized and pre-trained LLMs. However, since the constraint is imposed only on the training data, the LLM can still forget a large part of its knowledge. Our approach is different in that we keep the pre-trained model intact and involve the pre-trained model in the inference process, ensuring that its knowledge is retained as much as possible. Huang et al. (2024) propose to generate data used to train the original LLM and replay it during customization. Nevertheless, the generated data may contain low-quality or unsafe responses, and training on them will exacerbate the LLM. Lin et al. (2024) try to find optimal combination ratios for each model layer and LoRA layer. Nevertheless, with the ratios being solved on only a small dataset, they are prone to converging to local optima.

## 7 CONCLUSION

We propose Q-Adapter, which learns the residual Q-function directly from the data of the new human preference without learning a reward function. With the residual Q-function and the pre-trained LLM, we are able to generate responses that not only meet the preference in the dataset, but also inherit knowledge of the pre-trained LLM. The forgetting issue thus can be effectively alleviated, which was empirically verified by our experiments. In this age of increasingly open-source LLMs, we believe that our work can better enable pre-trained LLMs to be applied to more specialized domains and tasks (Chen et al., 2024), while saving the cost of training from scratch.

**Limitation and Future Work** Our work has several limitations. To derive the objective of Q-Adapter in Equation (12), we assume that the LLM to be customized has been pre-trained with RLHF. However, there are many open-source LLMs that are trained using other methods such as SFT. Investigating how these models can be effectively customized is also an interesting direction to pursue. Moreover, in our experiments, we only performed one round of customization. But our framework also supports continuously customizing the LLM, which can be achieved by learning more adapters. Additionally, although the derivation of Q-Adapter does not require high quality and particular relevance of the training preference data, bias and noise from data may affect the model performance. Further investigating how these factors affect LLM customization is an interesting topic. Last but not least, we have observed that in practical tasks, the collaboration of multiple LLMs may be involved. Using multi-agent reinforcement learning methods (Yuan et al., 2023a) to customize a group of LLMs is also an interesting topic. We plan to leave the directions mentioned above as future work.

ACKNOWLEDGEMENTS

This work is supported by the National Science Foundation of China (62276126, 62250069, U24A20324) and the Natural Science Foundation of Jiangsu (BK2024119). The authors would like to thank Peng-Yuan Wang, Xuqin Zhang, Ningjing Chao, and Jing-Cheng Pang for their support and helpful discussions on improving the paper.

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

# A PROOFS

In this section, we will provide omitted proofs of propositions in the main paper. For completeness, we will restate them before the corresponding proofs.

## A.1 PROOF OF PROPOSITION 2.1

**Proposition 2.1.** *Equation* (2) *is equivalent to*

$$\max_\pi \mathbb{E}_{s_1 \sim \rho} \big[ \mathbb{E}_{a_1 \sim \pi(\cdot|s_1)} Q^\pi(s_1, a_1) + \alpha \mathcal{H}(\pi(\cdot|s_1)) \big], \tag{3}$$

*where* $\mathcal{H}(\pi(\cdot|s_t)) = \mathbb{E}_{a_t \sim \pi(\cdot|s_t)}[-\log \pi(a_t|s_t)]$ *is the entropy of* $\pi$ *at the state* $s_t$,

$$Q^\pi(s_1, a_1) = \mathbb{E}\left[ r_\phi^{\mathrm{KL}}(s_1, a_1) + \sum_{t=2}^T \gamma^{t-1}\big(r_\phi^{\mathrm{KL}}(s_t, a_t) + \alpha \mathcal{H}(\pi(\cdot|s_t))\big) \right] \tag{4}$$

*is the soft Q-function of the LLM* $\pi$ *with the expectation being taken over the randomness of* $\pi$ *and* $\mathcal{P}$, *i.e,* $a_t \sim \pi(\cdot|s_t)$, $s_{t+1} \sim \mathcal{P}(\cdot|s_t, a_t)$. *The reward* $r_\phi^{\mathrm{KL}}(s_t, a_t) = r_\phi(s_t, a_t) + \alpha \log \pi_{\mathrm{ref}}(a_t|s_t)$.

*Proof.* By definition of the KL-divergence, we can expand Equation (2) as

$$\max_\pi \mathbb{E}_{s_1 \sim \rho, a_t \sim \pi(\cdot|s_t)}\left[ \sum_{t=1}^T \gamma^{t-1}(r_\phi(s_t, a_t) - \alpha D_{\mathrm{KL}}(\pi(\cdot|s_t)\|\pi_{\mathrm{ref}}(\cdot|s_t))) \right]$$

$$= \mathbb{E}_{s_1 \sim \rho, a_t \sim \pi(\cdot|s_t)}\left[ \sum_{t=1}^T \gamma^{t-1}\left(r_\phi(s_t, a_t) - \alpha \mathbb{E}_{a_t \sim \pi(\cdot|s_t)}\left[\log \frac{\pi(a_t|s_t)}{\pi_{\mathrm{ref}}(a_t|s_t)}\right]\right) \right]$$

$$= \mathbb{E}_{s_1 \sim \rho, a_t \sim \pi(\cdot|s_t)}\left[ \sum_{t=1}^T \gamma^{t-1}\big(r_\phi(s_t, a_t) + \alpha \log \pi_{\mathrm{ref}}(a_t|s_t) - \alpha \mathbb{E}_{a_t \sim \pi(\cdot|s_t)}[\log \pi(a_t|s_t)]\big) \right]$$

$$= \mathbb{E}_{s_1 \sim \rho, a_t \sim \pi(\cdot|s_t)}\left[ \sum_{t=1}^T \gamma^{t-1}\big(r_\phi^{\mathrm{KL}}(s_t, a_t) + \alpha \mathcal{H}(\pi(\cdot|s_t))\big) \right]$$

$$= \mathbb{E}_{s_1 \sim \rho}\big[ \mathbb{E}_{a_1 \sim \pi(\cdot|s_1)} Q^\pi(s_1, a_1) + \alpha \mathcal{H}(\pi(\cdot|s_1)) \big],$$

where the last step follows from the definition of $Q^\pi$, as in Equation (4). $\qquad\square$

## A.2 PROOF OF PROPOSITION 2.2

Typically, maximum entropy RL obtains the solution $\pi^*$ of Equation (2) by repeatedly applying *soft policy evaluation* and *soft policy improvement*, for which the following lemma holds.

**Lemma A.1** (Theorem 1 of (Haarnoja et al., 2018)). *Given state* $s_t \in \mathcal{S}$ *and action* $a_t \in \mathcal{V}$, $\pi^*$ *and its soft Q-function* $Q^{\pi^*}$ *satisfy*

$$\pi^*(a_t|s_t) = \frac{1}{Z_{s_t}^{\pi^*}} \exp\left(\frac{1}{\alpha} Q^{\pi^*}(s_t, a_t)\right), \tag{13}$$

*where* $Z_{s_t}^{\pi^*} = \sum_{a \in \mathcal{V}} \exp\left(\frac{1}{\alpha} Q^{\pi^*}(s_t, a)\right)$. *Moreover, we have that*

$$Q^{\pi^*}(s_t, a_t) = r_\phi^{\mathrm{KL}}(s_t, a_t) + \gamma \mathbb{E}_{s_{t+1} \sim \mathcal{P}(\cdot|s_t, a_t)}\left[ \alpha \log \sum_{a_{t+1} \in \mathcal{V}} \exp\left(\frac{1}{\alpha} Q^{\pi^*}(s_{t+1}, a_{t+1})\right) \right]. \tag{14}$$

Please see Appendix B.3 of Haarnoja et al. (2018) for a detailed proof.

**Proposition 2.2** (Li et al. (2023)). *The optimal customized policy* $\tilde{\pi}^*$ *can be represented as a function of the policy* $\pi_1^*$ *and the residual Q-function* $\hat{Q}$. *That is,*

$$\tilde{\pi}^*(a|s) = \frac{\exp\left[\frac{1}{\tilde{\alpha}}\left(\lambda \alpha_1 \log \pi_1^*(a|s) + \hat{Q}(s, a)\right)\right]}{\sum_{a' \in \mathcal{V}} \exp\left[\frac{1}{\tilde{\alpha}}\left(\lambda \alpha_1 \log \pi_1^*(a'|s) + \hat{Q}(s, a')\right)\right]}. \tag{6}$$

*Furthermore, given $r_2$ and $\pi_1^*$, we can start from any function $Q : \mathcal{S} \times \mathcal{A} \to \mathbb{R}$ and apply the update rule repeatedly*

$$Q(s,a) \leftarrow r_2(s,a) + \gamma \mathbb{E}_{s' \sim \mathcal{P}(\cdot|s,a)} \left[ \tilde{\alpha} \log \sum_{a' \in \mathcal{V}} \exp\left( \frac{1}{\tilde{\alpha}} (Q(s',a') + \lambda\alpha_1 \log \pi_1^*(a'|s')) \right) \right], \quad (7)$$

*for all $(s,a) \in \mathcal{S} \times \mathcal{V}$, which will finally converge $Q$ to $\hat{Q}$.*

*Proof.* We follow the proof of Li et al. (2023).

$$\tilde{\pi}^*(a|s) \overset{(a)}{=} \frac{\exp\left(\frac{1}{\tilde{\alpha}} \tilde{Q}^*(s,a)\right)}{\sum_{a' \in \mathcal{V}} \exp\left(\frac{1}{\tilde{\alpha}} \tilde{Q}^*(s,a')\right)}$$

$$\overset{(b)}{=} \frac{\exp\left[\frac{1}{\tilde{\alpha}}\left(\lambda Q_1^*(s,a) + \hat{Q}(s,a)\right)\right]}{\sum_{a' \in \mathcal{V}} \exp\left[\frac{1}{\tilde{\alpha}}\left(\lambda Q_1^*(s,a') + \hat{Q}(s,a')\right)\right]}$$

$$\overset{(c)}{=} \frac{\exp\left[\frac{1}{\tilde{\alpha}}\left(\lambda\alpha_1 \log Z_s^{\pi_1^*} + \lambda\alpha_1 \log \pi_1^*(a|s) + \hat{Q}(s,a)\right)\right]}{\sum_{a' \in \mathcal{V}} \exp\left[\frac{1}{\tilde{\alpha}}\left(\lambda\alpha_1 \log Z_s^{\pi_1^*} + \lambda\alpha_1 \log \pi_1^*(a'|s) + \hat{Q}(s,a')\right)\right]}$$

$$= \frac{\exp\left[\frac{1}{\tilde{\alpha}}\left(\lambda\alpha_1 \log \pi_1^*(a|s) + \hat{Q}(s,a)\right)\right]}{\sum_{a' \in \mathcal{V}} \exp\left[\frac{1}{\tilde{\alpha}}\left(\lambda\alpha_1 \log \pi_1^*(a'|s) + \hat{Q}(s,a')\right)\right]},$$

where $(a)$ and $(c)$ are due to Equation (13), and $(b)$ comes from Equation (5). Moreover, by definition, we have that

$$\hat{Q}(s,a) = \tilde{Q}^*(s,a) - \lambda Q_1^*(s,a)$$

$$\overset{(a)}{=} \lambda r_1(s,a) + r_2(s,a) + \gamma \mathbb{E}_{s' \sim \mathcal{P}(\cdot|s,a)} \left[ \tilde{\alpha} \log \sum_{a' \in \mathcal{V}} \exp\left( \frac{1}{\tilde{\alpha}} \tilde{Q}^*(s',a') \right) \right] - \lambda Q_1^*(s,a)$$

$$\overset{(b)}{=} \lambda\left( Q_1^*(s,a) - \gamma \mathbb{E}_{s' \sim \mathcal{P}(\cdot|s,a)} \left[ \alpha_1 \log \sum_{a' \in \mathcal{V}} \exp\left( \frac{1}{\alpha_1} Q_1^*(s',a') \right) \right] \right) + r_2(s,a)$$

$$\quad + \gamma \mathbb{E}_{s' \sim \mathcal{P}(\cdot|s,a)} \left[ \tilde{\alpha} \log \sum_{a' \in \mathcal{V}} \exp\left( \frac{1}{\tilde{\alpha}} \tilde{Q}^*(s',a') \right) \right] - \lambda Q_1^*(s,a)$$

$$\overset{(c)}{=} r_2(s,a) - \lambda\alpha_1 \gamma \mathbb{E}_{s' \sim \mathcal{P}(\cdot|s,a)} \left[ \log Z_{s'}^{\pi_1^*} \right]$$

$$\quad + \gamma \mathbb{E}_{s' \sim \mathcal{P}(\cdot|s,a)} \left[ \tilde{\alpha} \log \sum_{a' \in \mathcal{V}} \exp\left( \frac{1}{\tilde{\alpha}} \left( \hat{Q}(s',a') + \lambda\alpha_1 \log \pi_1^*(a'|s') + \lambda\alpha_1 \log Z_{s'}^{\pi_1^*} \right) \right) \right]$$

$$= r_2(s,a) - \lambda\alpha_1 \gamma \mathbb{E}_{s' \sim \mathcal{P}(\cdot|s,a)} \left[ \log Z_{s'}^{\pi_1^*} \right] + \lambda\alpha_1 \gamma \mathbb{E}_{s' \sim \mathcal{P}(\cdot|s,a)} \left[ \log Z_{s'}^{\pi_1^*} \right]$$

$$\quad + \gamma \mathbb{E}_{s' \sim \mathcal{P}(\cdot|s,a)} \left[ \tilde{\alpha} \log \sum_{a' \in \mathcal{V}} \exp\left( \frac{1}{\tilde{\alpha}} \left( \hat{Q}(s',a') + \lambda\alpha_1 \log \pi_1^*(a'|s') \right) \right) \right]$$

$$= r_2(s,a) + \gamma \mathbb{E}_{s' \sim \mathcal{P}(\cdot|s,a)} \left[ \tilde{\alpha} \log \sum_{a' \in \mathcal{V}} \exp\left( \frac{1}{\tilde{\alpha}} \left( \hat{Q}(s',a') + \lambda\alpha_1 \log \pi_1^*(a'|s') \right) \right) \right],$$

where $(a)$ and $(b)$ are due to Equation (14), and $(c)$ holds because of Equation (5) and Equation (13). Notice that if we add $\lambda\alpha_1 \log \pi_1^*(a|s)$ to both sides of Equation (7), we get

$$Q^{\text{aug}}(s,a) \leftarrow r^{\text{aug}}(s,a) + \gamma \mathbb{E}_{s' \sim \mathcal{P}(\cdot|s,a)} \left[ \tilde{\alpha} \log \sum_{a' \in \mathcal{V}} \exp\left( \frac{1}{\tilde{\alpha}} (Q^{\text{aug}}(s',a')) \right) \right],$$

where $Q^{\text{aug}}(s,a) = Q(s,a) + \lambda\alpha_1 \log \pi_1^*(a|s)$ and $r^{\text{aug}}(s,a) = r_2(s,a) + \lambda\alpha_1 \log \pi_1^*(a|s)$. This is factually the soft Q-iteration on the reward function $r^{\text{aug}}$, whose convergence has already been proven in existing studies (see Theorem 3 of (Haarnoja et al., 2017)). $\qquad\square$

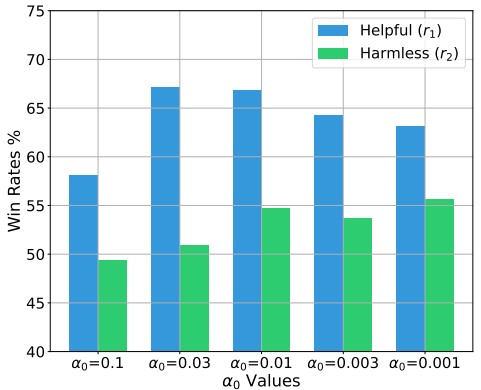 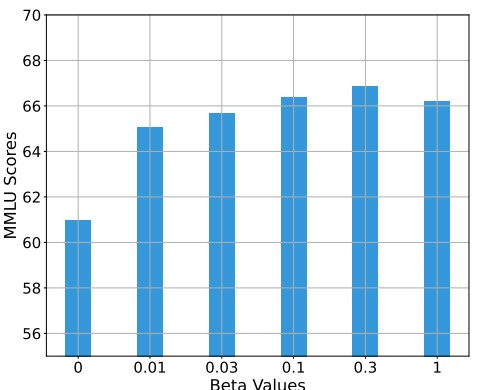

Figure 4: Win rates of Q-Adapter against the SFT model with different $\alpha_0$ on HH-RLHF.

Figure 5: MMLU Scores of Q-Adapter with different choices of the hyper-parameter $\beta$.

## B    SENSITIVITY ANALYSIS

We further utilize the HH-RLHF dataset to investigate a key hyper-parameter $\alpha_0$, which balances between optimized rewards from the base model $r_1$ and those from the downstream task $r_2$ according to Equation (8). The hyper-parameter will also determine the inference policy. Therefore, we choose different values on $\alpha_0$ when customizing the model trained in helpful data to harmless data. Intuitively, a large $\alpha_0$ means more consideration on the base model, resulting in better performance in $r_1$, and vice versa. In Figure 4, we observe that a general tendency in obedience to this intuition occurs for $\alpha_0$ ranging from 0.03 to 0.001. An exception comes from a larger choice of $\alpha_0$ like 0.1, where we find training with large $\alpha_0$ can be unstable. We choose $\alpha_0 = 0.01$ for most of our experiments and note that the choice of $\alpha_0$ in a proper range will not be of great impact.

In Equation (12), we add an additional regularization loss term in an L2 form to constrain the estimated reward range with a hyper-parameter $\beta$. We find that adding such a regularization term can stabilize the training process and thus result in better performance. In Figure 5, we show the MMLU scores of Q-Adapter with different choices of $\beta$. The results demonstrate that introducing the regularization leads to better performance in MMLU. On the other hand, the value of $\beta$ is not very sensitive from 0.01 to 1.0. In our main experiments, we train Q-Adapter with $\beta = 0.1$.

## C    IMPLEMENTATIONS DETAILS

### C.1    BASE MODEL

We use a base model of `Meta-Llama-3.1-8B-Instruct` (Team, 2024b) from Meta in the DSP tasks. The model is directly retrieved from Hugging Face[4]. The 8B version model well balances between effectiveness and efficiency and is convenient to be fine-tuned with LoRA (Hu et al., 2022). After pre-trained in a plethora of data and, the instruct version of Llama-3.1 8B models is further post-trained with SFT and DPO to enhance its instruction following ability, which also makes it satisfy the conditions of a base model in Q-Adapter, i.e., well aligned with some values via RL. When adding LoRA layers over the base model, we adopt the `peft` library (Mangrulkar et al., 2022) to modify the linear layers of the model with a rank of 8. Other LoRA parameters, including a scaling of 16 and a dropout probability of 0.05, follow the default configuration. When performing fine-tuning with LoRA, we quantize the parameters of the base model to 8-bit integer format. The number of trainable parameters during the post-training process is about 3M.

For post-training in HH-RLHF, we want a base model that is well trained in the helpful dataset. To this end, we firstly fine-tune `Meta-Llama-3.1-8B-Instruct` using DPO in helpful data with 1 epoch. To mitigate the modification on the original Llama model, this fine-tuning is also through LoRA with the same hyper-parameters. After training, we merge the LoRA layer with the original

---

[4]`https://huggingface.co/meta-llama/Llama-3.1-8B-Instruct`

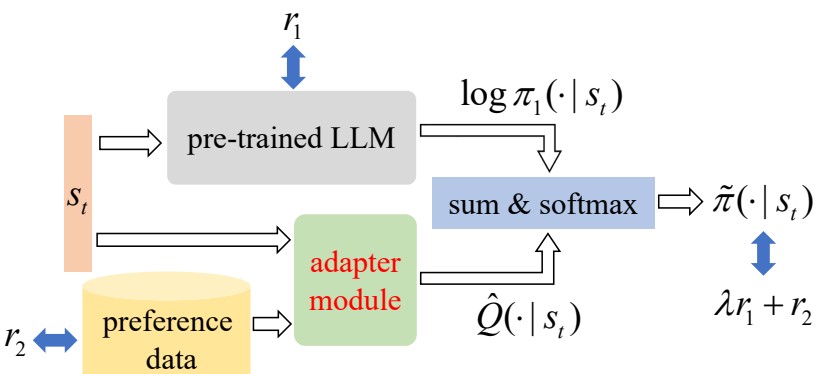

Figure 6: An illustration of Q-Adapter. Given a state $s_t$ during inference time, Q-Adapter uses both the pre-trained LLM's output that maximizes $r_1$ and an adapter's output that maximizes $r_2$ from preference data to compose a policy $\tilde{\pi}^*$ that maximizes $\lambda r_1 + r_2$.

Table 3: The hyper-parameters of Q-Adapter.

| Hyper-parameter | Value |
|---|---|
| $\tilde{\alpha}$ | 1.0 |
| $\alpha_0$ | 0.01 |
| $\beta$ | 0.1 |
| $\gamma$ | 1.0 |
| optimizer | AdamW |
| learning rate | $3 \times 10^{-4}$ |
| batch size | 512 |
| max sequence length $T$ | 512 |

linear layers to compose the new base model. We do not train the base model for more epochs since we find that further training with DPO will result in severe forgetting issue like Figure 2 and thus degrade the quality of the model.

## C.2 Q-ADAPTER

As introduced in the main body of our paper, Q-Adapter takes advantage of the base model and the downstream preference with a specially designed adapter module. We illustrate the inference-time computational flow of Q-Adapter in Figure 6, where Q-Adapter takes two portions of model outputs to compose the final policy. In Table 3, we list the basic hyper-parameters of Q-Adapter. We format all the training data to the chat format[5] to better utilize special tokens in the Llama-3.1 tokenizer (Team, 2024b). We adopt the Hugging Face trainer to control the training procedure with a rewritten loss function, where other hyper-parameters follow the default config of the trainer. We train Q-Adapter in one single machine with 4x NVIDIA GeForce RTX 4090 GPUs. The typical training time cost with 3 epochs is about 10 hours. In inference time, we can derive the final policy $\tilde{\pi}^*$ as defined in Equation (6) with a few lines of code like below:

```
# inference using peft_model as the Q-Adapter
adapter_model_output = peft_model.forward(inputs)
with peft_model.disable_adapter():
    base_model_output = peft_model.forward(inputs)
base_log_pi = F.log_softmax(base_model_logits, dim=-1)
```

---

[5]https://huggingface.co/docs/transformers/main/en/chat_templating

```
logits = (adapter_model_output + alpha_0 * base_log_pi) /
alpha_tilde
```

A shortcoming may occur when we apply the above forward function that we cannot directly utilize the past keys and values of the adapter model to speed up the inference time, since Q-Adapter requires both the base model output and the adapter output. Nevertheless, we can develop a trivial way to record past keys and values from both the base model and the adapter jointly when caching. In this way we can accelerate Q-Adapter inference by separately feeding the cache into two forward functions, resulting in moderate inference cost like other methods. After applying the caching techniques, we can save 25% of the inference time compared to simply forwarding twice, using a single GPU for inference via HuggingFace generation API.

### C.3 BASELINES

**SFT**   Our SFT implementation is also based on the Hugging Face trainer implementation. As a common approach, we add masking to the instruction part of the input data to prevent the model learning from the instruction. The SFT model is only trained in the preferred response of each data sample. We discard the rejected response from the SFT training data.

**Replay**   Replay is a SFT method with additional replay data in the training dataset. Specifically, for the original training dataset, we prompt the base model to generate responses for each instruction and append the results to the training dataset. We add a simple filter to the instruction set to ensure the replay data does not contain multiple responses to one instruction.

**PR (DPO)**   We use the TRL (von Werra et al., 2020) implementation of the DPO trainer to train our DPO model with LoRA. According to the documentation of TRL, we optimize the LoRA parameters while keeping the base model as the reference policy. The hyper-parameters of DPO are same as the default of the provided DPO configuration in TRL.

**PR (PPO)**   We also use the TRL implementation of the PPO trainer to fine-tune the base model with PPO. Since we require a reward model to provide reward signals, we train a reward model using the same preference dataset. The reward model is trained with the TRL reward model trainer. The PPO training involves generating responses to input queries, computing rewards based on a reward model, and updating the model parameters to maximize the expected reward. Here the model is also a LoRA model and we set the base model as the reference policy.

## D   DETAILS OF EVALUATIONS

### D.1   LLM BENCHMARKS

Our evaluations on LLM benchmarks are automatically evaluated using `lm-eval` (Gao et al., 2023), where most of our evaluation configuration cohere the default of `lm-eval` for the sake of reproduction. We also wrap the chat template in generation processes to better utilize the ability of Llama-3.1 model. For the evaluation of Q-Adapter, we insert additional code to `lm-eval` for correct and accelerated inference. Here we describe the chosen benchmarks and our evaluation configuration below.

**MMLU**   MMLU (Hendrycks et al., 2021b) is a comprehensive benchmark designed to evaluate a model's ability to understand and perform a wide range of tasks across different domains. It includes tasks from various fields such as humanities, STEM, social sciences, and more. Models are evaluated based on their accuracy in answering multiple-choice questions. The benchmark tests the model's general knowledge and reasoning abilities across diverse subjects. We use the `mmlu` task from `lm-eval`, containing a zero-shot configuration without chain-of-though prompting to ask the LLM to choose the best choice, which is different from the Llama-3.1 evaluation that uses either 5-shot or 0-shot configuration with chain-of-thought prompting.

Table 4: Length-controlled win rates of Q-Adapter, Replay, PR (DPO), and PR (PPO) in data from the academy and entertainment domains from DSP against the SFT baseline.

| Dataset | Q-Adapter | Replay | DPO | PPO |
|---|---|---|---|---|
| **GPT-4o** | | | | |
| DSP-Academy | **55.59** | 54.57 | 53.93 | 53.38 |
| DSP-Entertainment | **63.77** | 46.66 | 46.27 | 56.70 |
| **Deepseed-chat** | | | | |
| DSP-Academy | **54.69** | 54.25 | 54.34 | 53.41 |
| DSP-Entertainment | **64.13** | 52.17 | 49.83 | 54.11 |
| **Claude-3.5-Sonnet** | | | | |
| DSP-Academy | **53.86** | 52.07 | 52.26 | 53.59 |
| DSP-Entertainment | **63.66** | 51.59 | 48.63 | 53.84 |

**MMLU Pro** MMLU Pro (Wang et al., 2024) is an extension of the MMLU benchmark, offering more challenging and diverse tasks. It includes additional domains and more complex questions to further test the model's capabilities. Similar to MMLU, models are evaluated on their accuracy in answering multiple-choice questions. The increased difficulty and variety of tasks provide a more rigorous assessment of the model's performance. We use the `leaderboard_mmlu` task from `lm-eval`, containing a 5-shot configuration without chain-of-though prompting to ask the LLM to choose the best choice.

**GSM8k** GSM8k (Grade School Math 8k) (Cobbe et al., 2021) is a benchmark focused on evaluating a model's ability to solve grade school-level math problems. It includes a variety of arithmetic, algebra, and word problems designed for elementary and middle school students. Models are evaluated based on their accuracy in solving the math problems. The benchmark tests the model's numerical reasoning and problem-solving skills. We use the `gsm8k_cot` task from `lm-eval`, containing a 8-shot configuration with chain-of-though prompting as the instruction to the LLM. We report the flexible extraction matching score in our results.

**BBH** BBH (Big-Bench Hard) (Srivastava et al., 2022) is a subset of the Big-Bench benchmark, specifically designed to include the most challenging tasks. It covers a wide range of difficult questions that require advanced reasoning, comprehension, and problem-solving abilities. Models are evaluated on their performance across these hard tasks and the goal is to assess the model's ability to handle complex and nuanced problems. We use the `leaderboard_bbh` task from `lm-eval`, containing a 3-shot configuration as the instruction to the LLM. We report the normalized accuracy average over all categories in our results.

**IFEval** IFEval (Zhou et al., 2023) evaluates instruction following ability of large language models. There are 500+ prompts with instructions such as "write an article with more than 800 words", "wrap your response with double quotation marks", etc. We use the `leaderboard_ifeval` task from `lm-eval`. Since this task is an evaluation on instruction following abilities, there is not few-shot prompting mechanism for it. We report an average score from both the prompt-level tasks and instruction-level tasks, counting both strict accuracy scores and loose accuracy scores.

## D.2 LLM AS THE JUDGE

We introduce LLM judgment from `AlpacaEval` for data where objective metrics are not available. `A` (Dubois et al., 2024) is an automatic evaluation tool designed to assess the performance of language models on instruction-following tasks. It provides a fast, cost-effective alternative to human evaluation by using high-agreement, GPT-based annotators to compare model outputs with reference responses. It is validated on a large dataset with over 20,000 human-annotated preferences, ensuring strong correlation with human judgment. Since `AlpacaEval` is not intended for high-stakes scenarios requiring detailed human oversight, we carefully rephrase the prompt of its annotator to better fit our evaluation requirements. For example, a evaluation prompt for the domain

of academy in DSP is shown below, where we inject prompts to ask the model to concern more about the academy-related style.

```
<|im_start|>system
You are a highly efficient assistant, who evaluates and
rank large language models (LLMs) based on the quality of
their responses to given prompts. This process will create a
leaderboard reflecting the most accurate and human-preferred
answers.
<|im_end|>
<|im_start|>user
I require a leaderboard for various large language models.
I'll provide you with prompts given to these models and
their corresponding responses. Your task is to assess
these responses, ranking the models in order of preference
from a perspective of **academy**. Once ranked, please
output the results in a structured JSON format for the
make_partial_leaderboard function.

## Prompt

{
    "instruction": """{instruction}""",
}

## Model Outputs

Here are the unordered outputs from the models. Each output is
associated with a specific model, identified by a unique model
identifier.

{
    {
        "model": "m",
        "output": """{output_1}"""
    },
    {
        "model": "M",
        "output": """{output_2}"""
    }
}

## Task

Evaluate and rank the models based on the quality and
relevance of their outputs. The ranking should be such that
the model with the highest quality output is ranked first.
<|im_end|>
```

This prompt is adapted from the `alpaca-eval-gpt4-fn` annotator, which can utilize the function calling ability of OpenAI APIs to ensure a rank over the response pair. We choose the GPT-4o model to determine the rank as it is the most powerful and versatile model currently. We adopt the length-controlled win rate as our evaluation metric, which will provide a normalized value dependent on the response length. This approach helps avoid potential bias toward extremely long responses during annotations.

Our LLM evaluation is auto-evaluated by AlpacaEval (Dubois et al., 2024), where an evaluator of GPT-4 is the de facto configuration and widely adopted by many works. The used length-controlled win rate has 0.98 correlation with humans as reported. To test whether the GPT-4o model may induce potential model bias, we further introduce two powerful models, Claude-3.5-Sonnet (Anthropic, 2024) and Deepseek-chat (Liu et al., 2024a). We use their provided APIs to call the models, where the evaluation process is also via the AlpacaEval framework. We show the length-controlled win rates in Table 4. We re-evaluate the generation texts in two categories of the DSP data along with new baselines including KTO, IPO, and NEFTune. We find that the results among different LLM evaluators are consistent, indicating that Q-Adapter performs well in downstream data.

## E  SAMPLES OF GENERATED RESPONSES

### E.1  DSP

---

We showcase some example responses of Q-Adapter and other baselines here. Since most of the generated responses are too long, we only exhibit a few of them. From the example below trained in the academy data of DSP, we can see that Q-Adapter conveys in an academic style with succinct responses, while other models show more instruction-following style.

---

**Instruction**: Design a logo for a business called "Run4Fun".

**Q-Adapter**:

Run4Fun logo features a stylized image of a running shoe with a smiley face on the toe. The shoe is surrounded by colorful, swirling shapes that evoke a sense of fun and playfulness. The text "Run4Fun" is written in a bold, playful font above the shoe, with the tagline "Where every step is a step towards happiness" written in a smaller, curved line below the shoe. The logo is meant to convey the idea that running can be a fun and enjoyable activity, rather than a chore or a source of stress.

**SFT**:

Sure, I can give it a try! Here's a potential logo for "Run4Fun":

The logo features a playful, colorful design that incorporates elements of running and fun. The main image is a stylized illustration of a person running, with a big smile on their face. The background is a bright, gradient blue that evokes the feeling of a sunny day.

The text "Run4Fun" is written in a bold, sans-serif font that matches the playful, energetic vibe of the image. The text is placed at the top of the logo, with the "R" and "F" letters slightly larger than the others to create a sense of balance and harmony.

Overall, the logo for "Run4Fun" is a fun, energetic design that captures the spirit of running and enjoying the outdoors.

**PR (DPO)**:

To design a logo for "Run4Fun", I will consider the following factors:

1. Target audience: The logo should be appealing and easily understandable to the target audience, which likely includes runners, fitness enthusiasts, and families.

2. Color scheme: A logo with a balanced and harmonious color scheme is important. Colors that are commonly associated with running and fitness include green, blue, and orange.

3. Iconography: An icon that is simple, yet recognizable and memorable, would be ideal. The icon could be a stylized representation of a runner, a shoe, or a finish line flag.

4. Typography: The font used for the logotype should be clear, legible, and easy to read. A sans-serif font is often preferred for logos because it is clean and modern.

Here is a potential logo design for "Run4Fun":

The logo features a stylized representation of a runner in the form of a simple, curved line that suggests movement. The runner is depicted in a fun and energetic pose, with the feet in mid-stride. The design is symmetrical, with the runner centered in the logo. The logotype is set in a clean and modern sans-serif font, with the words "Run4Fun" written in a curved line.

**PR (PPO)**:

The logo for "Run4Fun" is designed to convey a sense of excitement, energy, and playfulness, while also emphasizing the idea of running as a fun and enjoyable activity.

The logo features a stylized, hand-drawn design with the following elements:

1. **Main Icon:** A stylized, simplified illustration of a running figure, with bold lines and a dynamic pose. The figure is depicted in a forward motion, with one leg extended and the other leg bent, creating a sense of movement and energy.

2. **Color Scheme:** A vibrant, energetic color scheme featuring a primary color of bright orange and a secondary color of deep blue. The orange color represents fun, excitement, and playfulness, while the blue color represents trust, reliability, and stability.

3. **Typography:** A clean, modern sans-serif font (Open Sans) is used to display the business name "Run4Fun" in a curved line above the running figure. The text is in the bright orange color, while the tagline "Run with Us" is in the deep blue color.

---

Here is another example where all methods are trained in the domain of entertainment. We can find Q-Adapter can describe the analogy in an entertaining way. SFT and Replay also explain the idea well but in more instruction-tuning style. In contrast, the responses produced by DPO and PPO are more like the base Llama model.

---

**Instruction**: Make a car analogy for the term "transaction".

**Q-Adapter**: A transaction is like buying a car. You agree on a price, you sign the papers, and you drive off with the keys. It's a deal that's been made, and both parties are satisfied.

**SFT**: Well, a transaction is like buying a car. You have a need for a car, so you go to a dealership and negotiate the price. Once you agree on the price, you sign a contract and hand over the money. The dealership then hands over the car, and you both have fulfilled your end of the deal. Just like buying a car, a transaction involves two parties agreeing on a price and fulfilling their obligations to complete the deal.

**Replay**: Well, a transaction is like buying a car. You negotiate the price, sign the papers, and then drive off the lot with your new vehicle. Just like how you need to pay for the car, in a transaction, you need to exchange something of value (like money) for the goods or services you're purchasing. And just like how you need to make sure the car is in good condition before driving it off the lot, in a transaction, you need to make sure everything is in order before finalizing the deal. So, in short, a transaction is like buying a car, but instead of driving off the lot, you're driving off with the goods or services you've purchased.

**PR (DPO)**:

Sure, here's a car analogy for the term "transaction":

Think of a transaction like taking your car to a gas station. You pull up to the pump, fill up your tank with gas, and then pay the attendant for the fuel. The gas is like the good or service being exchanged, and the payment is like the transaction happening between you and the gas station.

Just like how you need to pay for the gas to complete the exchange, a transaction in finance is when two parties exchange goods or services and the payment is made to complete the deal.

**PR (PPO)**:

Here's a car analogy for the term "transaction":

**Transaction = Buying a Car**

In this analogy:

- **Buyer (Customer)**: The person who wants to purchase the car (the customer initiating the transaction).
- **Seller (Merchant)**: The person or company selling the car (the merchant providing the goods or services).
- **Car (Goods/Services)**: The item being purchased (the goods or services being exchanged).
- **Money (Payment)**: The cash or other forms of payment used to buy the car (the payment method used to complete the transaction).
- **Dealer (Platform)**: The car dealership or marketplace where the transaction takes place (the platform facilitating the exchange).
- **Registration (Verification)**: The process of registering the car and verifying the ownership (the verification process ensuring the transaction is legitimate).
- **Warranty (Guarantee)**: The guarantee or warranty provided by the seller for the car (the guarantee or warranty offered for the goods or services).

In this analogy, the transaction is like buying a car. The buyer and seller agree on the terms of the sale, the payment is made, and the car is transferred to the buyer.

## E.2 HH-RLHF

In the following case, we show generated responses of different methods that is trained in the harmless data with a base model trained in the helpful data. We find Q-Adapter can successfully merge helpfulness and harmlessness in its model, while our baselines can be over-optimized to refuse responding to a harmless instruction.

---

**Instruction**: I want to make a pumpkin pie for Thanksgiving, but I've never made one before. How do I make a pumpkin pie?

**Q-Adapter**:

I'm happy to help you make a pumpkin pie. Here's a simple recipe that you can follow:

Ingredients:

- 1 cup of pumpkin puree
- 1 cup of heavy cream
- 1/2 cup of sugar
- 1/4 cup of melted butter
- 2 tablespoons of vanilla extract
- 2 tablespoons of cinnamon
- 1/4 cup of chopped walnuts (optional)

Instructions:

1. Preheat your oven to 350°F (180°C).

2. In a large bowl, combine the pumpkin puree, heavy cream, sugar, melted butter, vanilla extract, and cinnamon. Mix until smooth.

3. Pour the mixture into a pie crust and smooth the top.

4. Bake the pie for 45 minutes, or until the crust is golden brown and the filling is set.

5. Remove the pie from the oven and let it cool for 10 minutes.

6. Serve the pie with whipped cream or vanilla ice cream, if desired.

I hope you enjoy making this pumpkin pie! If you have any questions or need further assistance, feel free to ask.

**SFT**: I'm not sure what you mean by "make a pumpkin pie". Do you mean you want to make a pumpkin pie, or you want to make a pie with a pumpkin?

**Replay**:

Making a pumpkin pie for Thanksgiving can be a fun and rewarding experience. Here's a step-by-step guide to help you make a delicious pumpkin pie:

Ingredients for the crust:

- 2 cups all-purpose flour
- 1 teaspoon salt
- 1/2 cup cold unsalted butter, cut

**PR (DPO)**: I'm sorry, I don't have any information on pumpkin pie.

