# OpenReview forum: "Q-Adapter: Customizing Pre-trained LLMs to New Preferences with Forgetting Mitigation"
_ICLR.cc/2025/Conference — ICLR 2025 Poster_

### Official Review · Reviewer_16NH · 2024-11-01

**Soundness:** 3
**Presentation:** 3
**Contribution:** 2
**Rating:** 6
**Confidence:** 4

**Summary:**

Large Language Models (LLMs), trained on a large amount of corpus, have demonstrated remarkable abilities. However, it may not be sufficient to directly apply open-source LLMs like Llama to certain real-world scenarios, since most of them are trained for general purposes. Thus, the demands for customizing publicly available LLMs emerge. This paper introduces Q-Adapter, a method that directly learns the residual Q-function from new human preference data, bypassing the need to learn a reward function. With the residual Q-function and the pretrained LLM, Q-Adapter are able to generate responses that not only meet the preference in the dataset, but also inherit knowledge of the pre-trained LLM. The experimental results show that Q-Adapter on both retaining existing knowledge and learning new preferences.

**Strengths:**

1. The writing is generally clear and fluent.
2. The idea is well-motivated and easy-to-follow.
3. Based on the results provided in the paper, the Q-Adapter demonstrates good performance.

**Weaknesses:**

1. In Table 1, under the Category: Business, the PR (PPO) method shows a significantly lower performance compared to the other baselines (this is not as evident in other categories).
2. The study could include full parameter fine-tuning and additional baselines for comparison, such as other forgetting mitigation techniques or model fine-tuning strategies (e.g., NEFTune), to provide a more comprehensive evaluation of Q-Adapter's performance.

**Questions:**

Please refer to “Weakness”

---

> ### Author Response · Authors · 2024-11-21
> **Official Comment by Authors**
>
> > W1: In Table 1, under the Category: Business, the PR (PPO) method shows a significantly lower performance compared to the other baselines (this is not as evident in other categories).
>
> We used the PPO algorithm implemented by TRL [1] and adopted the same LoRA training config as other methods. The phenomenon is that PPO is extremely unstable in the business category of the dataset. We tried a few runs and the results shown in Table 1 are the best outcome we observed in the experiments. We speculate that the instability of online PPO [2, 3] may be exacerbated in our training settings. Improving the stability of PPO can be an important direction to enhance LLM customization.
>
> [1] Leandro von Werra, etc. TRL: Transformer Reinforcement Learning. https://github.com/huggingface/trl
>
> [2] Huang, S., Dossa, R. F. J., Raffin, A., Kanervisto, A., & Wang, W. (2022). The 37 implementation details of proximal policy optimization. *The ICLR Blog Track 2023*.
>
> [3] Zheng, R., Dou, S., Gao, S., Hua, Y., Shen, W., Wang, B., ... & Huang, X. (2023). Delve into PPO: Implementation matters for stable rlhf. In *NeurIPS 2023 Workshop on Instruction Tuning and Instruction Following*.
>
> > W2: The study could include full parameter fine-tuning and additional baselines for comparison, such as other forgetting mitigation techniques or model fine-tuning strategies (e.g., NEFTune), to provide a more comprehensive evaluation of Q-Adapter's performance.
>
> Thanks for your suggestion. We add the comparisons to NEFTune, along with KTO and IPO proposed by other reviewers. Please refer to **global_Q1** in our global comment above. Since our proposed Q-Adapter only learns parameters from the LoRA modules, we keep the same parameter size for all baselines in training and do not tune full parameters to ensure fair comparisons.

---

### Official Review · Reviewer_JfJc · 2024-11-02

**Soundness:** 3
**Presentation:** 3
**Contribution:** 2
**Rating:** 6
**Confidence:** 3

**Summary:**

This paper proposes a Q-learning-based customization approach for adapting pre-trained LLMs to new human preferences. The authors develop a method called Q-Adapter, which uses a residual Q-function to learn new preferences while preserving the LLM's original knowledge. They conduct a series of experiments to validate the effectiveness of Q-Adapter compared with baseline methods.

**Strengths:**

The idea of using a Q-learning-based customization approach to adapt LLMs to new human preferences without overwriting existing knowledge is innovative. Most existing work focuses on fine-tuning or prompt-based adaptation, but these approaches cannot effectively preserve the model’s original knowledge, often leading to catastrophic forgetting and limiting the adaptability of LLMs in learning new preferences while retaining prior abilities. This Q-Adapter method employs a residual Q-function to incrementally learn user-specific preferences, as well as successfully mitigate forgetting issues associated with traditional fine-tuning.

The mathematical derivations are thorough and presented. And the workflow is reasonable, as it integrates Q-learning with residual functions to balance learning new preferences while maintaining existing knowledge systematically. This approach improves model adaptability and further enhances stability in customization tasks across various user-specific requirements.

**Weaknesses:**

In the experiment, the authors do not compare their method with some SOTA methods other than PPO and DPO (See Questions). Also they only use GPT-4o as a judge (Judgments from LLMs part). Although it is common to use LLM-as-a-judge for similar tasks, the authors may consider using more models, especially those with more explainability or reasoning ability for their ranking and evaluation.

**Questions:**

Authors may consider referring to and comparing with:
https://arxiv.org/abs/2404.14723
https://arxiv.org/abs/2402.01306
https://arxiv.org/abs/2405.00675
https://proceedings.mlr.press/v238/gheshlaghi-azar24a.html

---

> ### Author Response · Authors · 2024-11-21
> **Official Comment by Authors**
>
> > W1: Also they only use GPT-4o as a judge (Judgments from LLMs part). Although it is common to use LLM-as-a-judge for similar tasks, the authors may consider using more models, especially those with more explainability or reasoning ability for their ranking and evaluation.
>
> Thanks for your suggestion. We introduce two more powerful LLMs, Claude-3.5-Sonnet and Deepseek-chat, to evaluate generation qualities. We present the results in **global_Q2** from our general comment above, showing that results among these models are relatively consistent. Note that we only adopt LLM judgments when objective metric is inaccessible, particularly in evaluating generation qualities from downstream test data.
>
> > Q1: In the experiment, the authors do not compare their method with some SOTA methods other than PPO and DPO (See Questions); Authors may consider referring to and comparing with: https://arxiv.org/abs/2404.14723 https://arxiv.org/abs/2402.01306 https://arxiv.org/abs/2405.00675 https://proceedings.mlr.press/v238/gheshlaghi-azar24a.html
>
> We have added three baselines, KTO [1], IPO [2], and NEFTune [3], where KTO and IPO are policy regularization methods mentioned above and NEFTune are an enhanced SFT method mentioned by Reviewer 16NH. Please refer to **global_Q1** for the new results, from which we find that our method, Q-Adapter can outperforms these methods.
>
> [1] Ethayarajh, K., Xu, W., Muennighoff, N., Jurafsky, D., & Kiela, D. (2024). KTO: Model alignment as prospect theoretic optimization. *arXiv preprint arXiv:2402.01306*.
>
> [2] Azar, M. G., Guo, Z. D., Piot, B., Munos, R., Rowland, M., Valko, M., & Calandriello, D. (2024). A general theoretical paradigm to understand learning from human preferences. In *International Conference on Artificial Intelligence and Statistics* (pp. 4447-4455).
>
> [3] Jain, N., Chiang, P. Y., Wen, Y., Kirchenbauer, J., Chu, H. M., Somepalli, G., ... & Goldstein, T. (2023). NEFTune: Noisy embeddings improve instruction finetuning. *arXiv preprint arXiv:2310.05914*.

---

> > ### Comment · Reviewer_JfJc · 2024-11-22
> >
> > Thank you for explaining. I am looking forward to seeing the updated version.

---

> > > ### Author Response · Authors · 2024-11-22
> > >
> > > Thank you for your kind response! The updated version of our paper was already available at the time we posted the rebuttal. You can access the revised PDF by clicking the button at the top of the OpenReview page. We have highlighted the major revisions in magenta for your convenience. Please feel free to review the updates and let us know if our rebuttal addresses your concerns.

---

### Official Review · Reviewer_BxCH · 2024-11-04

**Soundness:** 3
**Presentation:** 3
**Contribution:** 2
**Rating:** 6
**Confidence:** 3

**Summary:**

This paper proposes a Q-learning-based customization approach for adapting LLMs to new user preferences. The authors introduce Q-Adapter, which employs residual Q-learning to incorporate specific preferences while retaining the LLM's original knowledge base. Experimental results show that Q-Adapter effectively mitigates forgetting, allowing customized models to perform well on both general and preference-specific tasks​.

**Strengths:**

1. The methods are described systematically and in detail. In the methodology section, the authors clearly explain each step in the Q-learning-based customization process.

2. The design choices, such as selecting embedding models, are well justified within the context of LLMs adaptation. The model compatibility and customization needs are clearly discussed, and the chosen embedding models align with the goals. This method effectively balances retaining general knowledge with integrating user-specific preferences.

3. Their clear discussion of each step of their methodology and the discussion of their variants prepare the reader for their experiments with the techniques.

**Weaknesses:**

1. In Section Judgments from LLMs, the evaluation with GPT-4o is not convincing. The LLM-as-a-judge is intended to provide an objective and robust assessment of model outputs across different dimensions. However, simply using GPT-4o for evaluation actually introduces potential biases and lacks the transparency needed for reliable performance validation.

2. For the experimental result, the authors may add more analysis as the performance of this Q-Adapter does not significantly outperform baseline methods. For example, there is minimal improvement in the BBH dataset (Table 1), and without error analysis, it is challenging to demonstrate the method's effectiveness.

3. Also the paper lacks a thorough discussion on the computational overhead or practical considerations associated with deploying Q-Adapter.

**Questions:**

1. The result in Table 1 is confusing, as the base model seems to have the best performance. Could any training harm the model's performance?

2. The paper employs the DSP and HH-RLHF datasets but does not explain the criteria for selecting these datasets over other domain-specific or preference-aligned datasets (Section 5.1, Experimental Setup). What are the key attributes of these datasets that make them particularly suitable for evaluating Q-Adapter’s capabilities in preserving and adapting model preferences?

---

> ### Author Response · Authors · 2024-11-21
> **Official Comment by Authors**
>
> > W1: In Section Judgments from LLMs, the evaluation with GPT-4o is not convincing. The LLM-as-a-judge is intended to provide an objective and robust assessment of model outputs across different dimensions. However, simply using GPT-4o for evaluation actually introduces potential biases and lacks the transparency needed for reliable performance validation.
>
> Thanks for your suggestion. As stated in our experiment setup, we ONLY use LLM judgments when there is NO objective metrics to quantify the generation qualities in downstream test data. When it is applicable to use objective metrics (i.e., measuring anti-forgetting capabilities), we have adopted them in general benchmarks in Table 1, including widely used assessments in MMLU, BBH, GSM8k and so on. Moreover, to ablate potential bias of GPT-4o, we add two more evaluators, Claude-3.5-Sonnet and Deepseek-chat, to evaluate generation texts from different methods. We present the results in **global_Q2** from the global comment above, where these evaluators show relatively consistent results.
>
> > W2: For the experimental result, the authors may add more analysis as the performance of this Q-Adapter does not significantly outperform baseline methods. For example, there is minimal improvement in the BBH dataset (Table 1), and without error analysis, it is challenging to demonstrate the method's effectiveness.
>
> We use Table 1 to measure the anti-forgetting capability of different methods during customization. In general, our method achieves the highest scores in 19 out of 20 comparisons from five benchmarks in four different datasets. To further describe the degree of forgetting, we find that the average score decrease of Q-Adapter is 30.0% less than the best baseline (Replay) and 43.2% less than the average performance of all baselines in Table 1, indicating the advantage of our method.
>
> > W3: Also the paper lacks a thorough discussion on the computational overhead or practical considerations associated with deploying Q-Adapter.
>
> We state the training cost of Q-Adapter in Appendix B.2 in our original paper. We can train Q-Adapter for 3 epochs in a single machine with 4x NVIDIA GeForce RTX 4090 GPUs. In inference time, the 8B version of Q-Adapter can be deployed in a single NVIDIA GeForce RTX 4090 GPU. Although Q-Adapter requires both the outputs from the base and the adapter model, we propose a caching workaround, which can save 25% the inference time compared to simply forwarding twice. We have updated Appendix B.2 to include the discussion. Thanks for your comment.
>
> > Q1: The result in Table 1 is confusing, as the base model seems to have the best performance. Could any training harm the model's performance?
>
> It is potential to harm the model’s general benchmark scores when we train the model in a different domain-specific data distribution. As stated in W2, the results are expected since all the methods suffer from the forgetting issue when we tune a model in a specific downstream dataset, which is our customization process intending to address. Please refer to **global_Q3** for further clarifications.
>
> > Q2: The paper employs the DSP and HH-RLHF datasets but does not explain the criteria for selecting these datasets over other domain-specific or preference-aligned datasets (Section 5.1, Experimental Setup). What are the key attributes of these  datasets that make them particularly suitable for evaluating Q-Adapter’s capabilities in preserving and adapting model preferences?
>
> In our original submission, we present details on why we choose DSP and HH-RLHF at the beginning of Sections 5.2 and 5.3, respectively. In Section 5.1 of the revised PDF, we have added a new paragraph on the reasons for choosing them to conduct experiments. In short, we do not deliberately select datasets that will give our method an advantage over the baselines. The reasons for choosing DSP and HH-RLHF are simply that 1) they are suitable for our considered customization scenarios and 2) they allow us to easily define $r_1$ and $r_2$, thereby facilitating subsequent evaluations.

---

> > ### Comment · Reviewer_BxCH · 2024-11-27
> >
> > Thanks for the detailed feedback. I will keep my rating, as I believe it is a good paper that should be accepted.

---

### Official Review · Reviewer_sDhN · 2024-11-04

**Soundness:** 3
**Presentation:** 3
**Contribution:** 3
**Rating:** 6
**Confidence:** 4

**Summary:**

The authors of this paper aim to discuss the customization process of LLMs, focusing on adding new preferences to open-source LLMs without losing original knowledge to meet specific application needs. The authors propose a novel approach called Q-Adapter. Q-Adapter redefines LLM customization as a reward maximization problem and utilizes a residual Q-learning framework with a new loss function to implement a customization process that aligns with new preferences. This framework enables direct learning on preference data, thereby avoiding the problem of forgetting the original model.

**Strengths:**

1. The idea of applying the reinforcement learning method to LLM customization is novel. Many existing works focus on adapting models to new contexts through fine-tuning. This can alter the model's original knowledge structure and decrease performance on other tasks. The method proposed by the authors enables alignment with new preferences without the need to change the original structure of the LLM.

2. Q-Adapter can learn directly from preference data without accessing or modifying the original reward functions used during the initial model training. This direct learning approach reduces the complexities of retraining or fine-tuning models on new datasets, making it easier to transfer and deploy the method to new tasks.

3. The authors provide sufficient mathematical derivations and experimental parameters in the paper and its appendices, which aids in replicating the experiments described in the paper.

**Weaknesses:**

1. According to Section 4, the effectiveness of this framework highly depends on the quality and relevance of the training preference data. The author may further discuss the impact of noise and the limitation of the model's ability to align with user preferences accurately within more datasets.

2. In Section 5, the authors only compare the Q-Adapter with two Policy Regularization baselines, DPO and PPO. The authors should include more up-to-date baselines in their experiments to better validate the effectiveness of their method.

**Questions:**

1. According to Section 5.1, the evaluation metrics may need further clarification. It seems that in all tasks, including MMLU, MMLU Pro, GSM8K, BBH, and IFEval, the Base Model shows the highest score compared to other methods.

2. In section 5.2, the authors evaluate the model's capabilities in learning new preferences only by demonstrating the evaluation of OpenAI GPT-4o. This evaluation is insufficient; the author may further discuss the evaluation results from other LLMs and human labelers.

---

> ### Author Response · Authors · 2024-11-21
> **Official Comment by Authors**
>
> > W1: According to Section 4, the effectiveness of this framework highly depends on the quality and relevance of the training preference data. The author may further discuss the impact of noise and the limitation of the model's ability to align with user preferences accurately within more datasets.
>
> We do not assume the quality and particular relevance of the training preference data in Section 4. These factors on data and noise may affect the performance of both our method and our baselines. In fact, the dataset we used is not perfect. For instance, the HH-RLHF dataset is shown to contain noise [1], but the experimental results indicate that our method outperforms the baselines. Further investigating how noise or model capability affects LLM customization is an interesting topic though not falling to our main contributions. We have revised our paper to add the discussion in Section 7.
>
> [1] Yeh, M. H., Tao, L., Wang, J., Du, X., & Li, Y. (2024). How Reliable Is Human Feedback For Aligning Large Language Models?. arXiv preprint arXiv:2410.01957.
>
> > W2: In Section 5, the authors only compare the Q-Adapter with two Policy Regularization baselines, DPO and PPO. The authors should include more up-to-date baselines in their experiments to better validate the effectiveness of their method.
>
> We originally choose DPO and PPO as the representative methods which adopt policy regularization to mitigate forgetting. To include more up-to-date approaches, we have added three baselines including KTO, IPO and NEFTune. Please refer to **global_Q1** and **global_Q2** in the global comment above for the results, where Q-Adapter still outperforms these baselines.
>
> > Q1: According to Section 5.1, the evaluation metrics may need further clarification. It seems that in all tasks, including MMLU, MMLU Pro, GSM8K, BBH, and IFEval, the Base Model shows the highest score compared to other methods.
>
> Our evaluation metrics in general benchmarks, including MMLU, MMLU Pro, GSM8K, BBH, and IFEval, is to measure the forgetting issue when we fine-tune models in downstream data. The performance drop is expected since our customization process uses a preference dataset with different traits from the math, reasoning, or instruction following capabilities indicated by the benchmarks above. In general, our method provides a better trade-off between general capabilities and downstream performance. We present a further clarification in **global_Q3**.
>
> > Q2: In Section 5.2, the authors evaluate the model's capabilities in learning new preferences only by demonstrating the evaluation of OpenAI GPT-4o. This evaluation is insufficient; the author may further discuss the evaluation results from other LLMs and human labelers.
>
> Thanks for your suggestion. Although the evaluator of GPT-4o can contain bias when make judgments, the auto-evaluation process of AlpacaEval along with the length-controlled technique can highly correlate to humans. In global_Q2, we present the evaluation results using two powerful LLMs, Claude-3.5-Sonnet and Deepseek-chat. We find these models provide relatively consistent results and Q-Adapter still maintain good performance.

---

> ### Comment · Reviewer_sDhN · 2024-11-27
>
> Thank you for the clarification; I will keep my score.

---

### Author Response · Authors · 2024-11-21
**Official Comment By Authors**

We sincerely thank all the reviewers for your time and efforts in reviewing our paper. In this global response, we would like to first summarize our revisions (highlighted in `magenta` color in the revised PDF). Clarifications to several concerns commonly raised by reviewers are then followed.

## Revisions

+ Add explanations on selecting DSP and HH-RLHF for experiments (Section 5.1).
+ Add analysis on why the Base Model performs the best in Table 1 (Section 5.2).
+ Add evaluations from Claude-3.5-Sonnet and Deepseek-Chat (Section 5.2 and Appendix C).
+ Add a concise discussion about the impact of noise on model performance (Section 7).
+ Add details on the computation requirements for deploying Q-Adapter (Appendix B.2).

## Global Responses

> **global_Q1**: Comparison with More Baselines (W2 of Reviewer sDhN, W1 of Reviewer BxCH, Q1 of Reviewer JfJc, W2 of Reviewer 16NH)

We have added more baselines including IPO, KTO, and NEFTune. IPO and KTO are two methods that adopt policy regularization to mitigate forgetting similar to DPO, while NEFTune is an enhanced SFT approach. We show the anti-forgetting capabilities of new baselines and Q-Adapter in the following table. The results demonstrate that Q-Adapter maintains the best performance among all methods. We find KTO generally shows similar performance to previous policy regularization baselines like DPO, and NEFTune can achieve slightly better performance than SFT. We also evaluate the performance of these additional baselines in downstream data through LLM judgments and put the results in the next question.

| Category | Method | MMLU | MMLU Pro | GSM8k | BBH | IFEval |
| --- | --- | --- | --- | --- | --- | --- |
| DSP-Academy | Q-Adapter | **66.40** | **36.37** | **77.86** | **47.34** | **68.14** |
|  | IPO | 62.55 | 34.46 | 73.39 | 45.13 | 50.19 |
|  | KTO | 65.31 | 33.27 | 74.33 | 44.56 | 66.60 |
|  | NEFTune | 64.67 | 34.29 | 74.68 | 44.87 | 60.95 |
| DSP-Entertainment | Q-Adapter | **66.69** | **35.98** | **77.94** | **46.33** | **69.15** |
|  | IPO | 63.09 | 32.65 | 72.55 | 40.04 | 55.71 |
|  | KTO | 65.00 | 33.92 | 75.66 | 45.15 | 65.44 |
|  | NEFTune | 63.77 | 31.19 | 74.00 | 44.63 | 58.50 |

> **global_Q2**: Evaluation with Other LLMs (Q2 of Reviewer sDhN, W1 from Reviewer BxCH, W1 of Reviewer JfJc)

Our LLM evaluation is auto-evaluated by AlpacaEval, where an evaluator of GPT-4 is the default configuration and wildly adopted by many works. The used length controlled win rate has 0.98 correlation with humans as reported by [1]. As the reviewers proposed to use other models for evaluation, we further introduce two LLMs, Claude-3.5-Sonnet and Deepseek-chat. We use their provided APIs to call the models, where the evaluation process is also via the AlpacaEval framework. We show the length-controlled win rates as follows.

| GPT-4o | Q-Adapter | Replay | DPO | PPO | IPO | KTO | NEFTune |
| --- | --- | --- | --- | --- | --- | --- | --- |
| DSP-Academy | **55.59** | 54.57 | 53.93 | 53.38 | 51.64 | 55.07 | 52.11 |
| DSP-Entertainment | **63.77** | 46.66 | 46.27 | 56.7 | 49.37 | 50.27 | 55.75 |

| Claude-3.5-Sonnet | Q-Adapter | Replay | DPO | PPO | IPO | KTO | NEFTune |
| --- | --- | --- | --- | --- | --- | --- | --- |
| DSP-Academy | **53.86** | 52.07 | 52.26 | 53.59 | 52.75 | 53.65 | 52.79 |
| DSP-Entertainment | **63.66** | 51.59 | 48.63 | 53.84 | 51.58 | 48.07 | 54.54 |

| Deepseed-chat | Q-Adapter | Replay | DPO | PPO | IPO | KTO | NEFTune |
| --- | --- | --- | --- | --- | --- | --- | --- |
| DSP-Academy | **54.69** | 54.25 | 54.34 | 53.41 | 54.9 | 54.85 | 53.95 |
| DSP-Entertainment | **64.13** | 52.17 | 49.83 | 54.11 | 50.02 | 44.61 | 53.02 |

We re-evaluate the generation texts in two categories of the DSP data along with new baselines including KTO, IPO, and NEFTune. We find that the results among different LLM evaluators are consistent, indicating that Q-Adapter performs well in downstream data.

[1] Dubois, Y., Galambosi, B., Liang, P., & Hashimoto, T. B. (2024). Length-controlled AlpacaEval: A simple way to debias automatic evaluators. *arXiv preprint arXiv:2404.04475*.

> **global_Q3**: Why does the base model perform the best in Table 1? (Q1 of Reviewer sDhN, Q1 of Reviewer BxCH)

It is expected when we customize LLMs in downstream data that has large discrepancies with these general benchmark. As shown in Figure 2, the forgetting issue clearly occurs in general benchmarks like MMLU, similar to previous findings [1]. Therefore, the major motivation of Q-Adapter is to investigate how to mitigate such forgetting. In general, the average decrease in benchmark scores of Q-Adapter is 30.0% less than the best baseline (Replay) and 43.2% less than the average performance of all baselines in Table 1, indicating the anti-forgetting capability of Q-Adapter during customization.

[1] Lin, Y., Lin, H., Xiong, W., Diao, S., Liu, J., Zhang, J., ... & Zhang, T. (2024). Mitigating the alignment tax of RLHF. In EMNLP.

---

### Author Response · Authors · 2024-11-25
**Kindly Request for Feedback**

Hi, reviewers! As the discussion period comes to an end, we want to follow up to see if the response addresses your concerns. If you have any further questions, please let us know. Thank you again!

---

### Meta-Review · Area_Chair_fKe5 · 2024-12-22

**Metareview:**

Summary: This paper introduces Q-Adapter, a novel Q-learning-based approach for customizing Large Language Models (LLMs) to align with new user preferences without overwriting their original knowledge. The method employs a residual Q-function to learn from preference data, enhancing model adaptability and stability across various user-specific requirements. The experimental results demonstrate Q-Adapter's effectiveness in retaining existing knowledge while learning new preferences.

Strengths:

Q-Adapter innovatively applies reinforcement learning to LLM customization, allowing for the integration of new preferences without compromising the model's original knowledge base.

The method provides a systematic balance between retaining general knowledge and adapting to specific user preferences, improving model versatility.

All the reviewers give the positive feedbacks， thus I recommend the acceptance of this work.

**Additional Comments On Reviewer Discussion:**

All the reviewers' concerns have been solved.

---

### Decision · Program_Chairs · 2025-01-22

Accept (Poster)